# The Complexity Dynamics of Grokking

## Abstract

We investigate the phenomenon of generalization through the lens of compression. In particular, we study the complexity dynamics of neural networks to explain *grokking*, where networks suddenly transition from memorizing to generalizing solutions long after over-fitting the training data. To this end we introduce a new measure of intrinsic complexity for neural networks based on the theory of Kolmogorov complexity. Tracking this metric throughout network training, we find a consistent pattern in training dynamics, consisting of a rise and fall in complexity. We demonstrate that this corresponds to memorization followed by generalization. Based on insights from rate–distortion theory and the minimum description length principle, we lay out a principled approach to lossy compression of neural networks, and connect our complexity measure to explicit generalization bounds. Based on a careful analysis of information capacity in neural networks, we propose a new regularization method which encourages networks towards low-rank representations by penalizing their spectral entropy, and find that our regularizer outperforms baselines in total compression of the dataset.

## 1 Introduction

A central aim of machine learning is to learn a *model* which *explains* a dataset. A good explanation—and therefore a good model—is one which *generalizes* to unseen data. How do we choose a good explanation? Occam's Razor says that all else being equal, less complex (i.e. simpler) explanations are expected to generalize better. This leads us to a fundamental open question in machine learning: how do we measure the complexity of a model that has been learned to explain a dataset?

In this paper, we make this question sharp, then answer it using the tools of information theory, compression, and the Minimum Description Length principle (MDL), which we apply to neural networks to explain grokking. The MDL principle (Rissanen, 1978) is based on the following observation: any regularity in a dataset can be used to compress that dataset. By Shannon's source coding theorem, the expected optimal coding rate for samples drawn from a distribution is equal to the entropy of the distribution. To improve the encoding, we decrease the apparent entropy of the data by finding patterns in it, which is the purpose of a model.

However, the specification of the model also requires information, so the total information is the sum of information used to specify the model, plus the remaining entropy of the dataset under the model. We have successfully compressed the data only if the total description of the model and the remaining entropy of the data is less than the naïve entropy under a random predictor. MDL tells us to minimize the sum of model complexity, denoted $C(M)$, and data entropy under the model, $H(D \mid M)$:

$$\min_M H(D \mid M) + C(M) \tag{1}$$

Note that if the model $M$ fully explains some subset $D_M \subseteq D$ then the prediction of that data becomes deterministic and $H(D_M) = 0$. Consider a trivial model which simply memorizes the data in a lookup table. Whilst this reduces the entropy of the data to zero, the model complexity is exactly equal to the original entropy. Therefore total description length has not changed, and no compression has been achieved by the model. The intuition that memorizing models are not expected to generalize is made formal via generalization bounds, which we discuss and relate to compression in section 3.

To avoid learning trivial memorizing models, practitioners introduce regularizers to the model training process – typically modified objective functions which encourage the model towards lower complexity solutions. Across a range of fields of statistical machine learning, generalization bounds of the form

$$\text{test error} \leq \text{train error} + \text{model complexity} \tag{2}$$

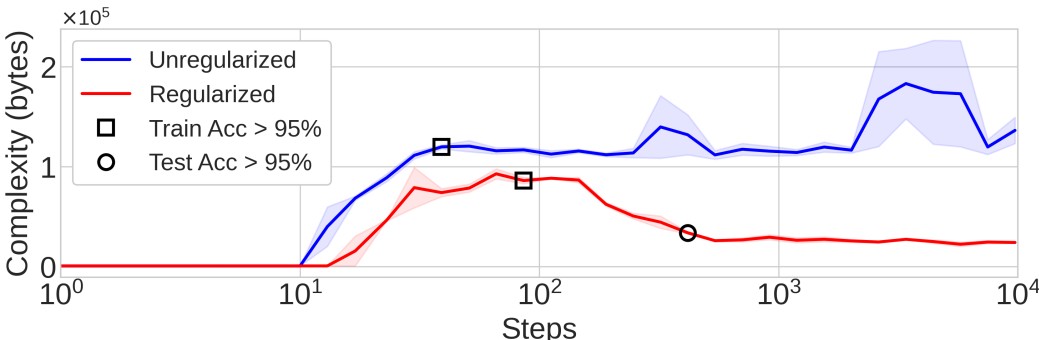

Figure 1: Complexity dynamics for unregularized (blue) and regularized (red) neural networks in grokking experiments on modular multiplication. The markers denote when memorization ($\square$) and generalization ($\bigcirc$) occur. In the regularized network, generalization occurs as complexity falls after its peak at memorization, while the unregularized network never generalizes, and its complexity remains large. We plot the mean over 3 seeds, shading std. error.

are ubiquitous (Vapnik, 1991; Bartlett & Mendelson, 2003). Notice the similar form of this bound and the MDL principle: if the error term is identified as the entropy of the data under the model, as is typically the case, then both MDL and the generalization bound are minimized by the sum of model complexity and data entropy under the model. That is, minimizers of the MDL principle naturally coincide with minimizers of generalization bounds.

To apply the MDL principle we require a computable measure of model complexity, $C(M)$. A widespread assumption in machine learning is that a model's *capacity* is a good approximation of its complexity. However, capacity is the maximum *upper-bound* on model complexity. A lower bound on complexity is given by the Kolmogorov complexity. Formally, the Kolmogorov complexity $K$ of a string $s$ is equal to the length of the shortest program $p$ that prints the string:

$$K(s) = \min_p \{\texttt{len}(p) : \texttt{exec}(p) = s\} \tag{3}$$

To link this back to the memorising model above, consider the string "11111..." composed of $N$ "1"s. A program which prints this string is: `print 1 N times`. The length of this program compared to the string's length is asymptotically small, hence the Kolmogorov complexity of the string is small. Next consider a random string of length $N$: "10110...", the shortest program which produces it hard-codes the string: `print "10110..."`. The length of this program is approximately equal to the length of the string. In fact, this is the definition of random in algorithmic information theory (AIT): a random string is one which has no description shorter than itself. Equivalently, random data is *incompressible*, since the shortest description of a random string is at least as long as that string.

While Kolmogorov complexity is uncomputable, compression provides an upper bound: the compressed data plus its decompressor form a program that generates the original data. We exploit this fact to produce complexity estimates via compression bounds, and outperform naïve compressors like `bzip2` by over an order of magnitude. The sensitivity to the underlying complexity dynamics we achieve is crucial, and enables us to distinguish different complexity phases of the grokking phenomenon, as shown in Fig. 1. Plotting naïve compression of networks in Fig 1 would show a flat line at an order of magnitude larger complexity, and would fail to meaningfully track complexity dynamics in the network. Our method achieves compression ratios of 30–40× over `bzip2`.

Our central theoretical contribution is a principled approach to lossy compression of neural networks, which is best understood as a formalization of the "apparent complexity" measure introduced by Aaronson et al. (2014) using the language of algorithmic information and rate–distortion theory. We apply the resulting compression-based complexity bound to track the complexity dynamics of neural networks as they transition from memorization to generalization. The most striking example of this transition is the recently observed grokking phenomenon (Power et al., 2022), where networks suddenly transition from memorization to generalization long after over-fitting the training data.

Our main contributions are the following:

- We introduce a new measure of complexity for neural networks, based on Kolmogorov complexity and rate–distortion theory. Our complexity measure is best understood as a principled lossy compression scheme for neural networks, and outperforms naïve compression baselines by 30–40×. Our measure enables us to sensitively track the complexity dynamics of neural networks throughout training.

- We explain grokking by tracking the complexity dynamics of neural networks as they transition from memorization to generalization. We find that properly regularized networks exhibit a characteristic rise and fall of complexity, while models which over-fit remain highly complex after memorizing the training data.

- We propose a regulariztion scheme which encourages networks towards low complexity by penalizing their *spectral entropy*, a differentiable measure of the intrinsic dimension of the network. We compare the complexity dynamics of different regularizers, and find that our regularizer results in the least complex model amongst the baselines we study.

## 2 RELATED WORK

**Grokking**    The grokking phenomenon (Power et al., 2022) occurs when networks suddenly transition from memorizing to generalizing solutions long after over-fitting the training data. Liu et al. (2022a) characterize conditions under which grokking occurs, and categorize learning into four distinct regimes, *comprehension*, *grokking*, *memorization*, and *confusion* by examining phase diagrams of learning performance across different hyperparameters. Liu et al. (2022b) demonstrate grokking on non-algorithmic datasets, and propose a link between a network's parameter norm and grokking. This link is also explored by Varma et al. (2023), who explain grokking in terms of the "efficiency" of a network, as measured by the ratio of the logit to parameter norms. Nanda et al. (2023) propose task-specific loss-based "progress measures" to detect grokking, but note that we "lack a general notion of criticality that would allow us to predict when the phase transition" occurs. In this work, we suggest that complexity is the key dynamical metric to understand grokking (and generalization more broadly), and discuss why proxy measures such as the $L^2$ norm are insufficient.

**Generalization**    The connection between model capacity and complexity is sometimes misunderstood amongst machine learning practitioners: the contemporary view in deep learning is that "larger models are better" (Nakkiran et al., 2019), with empirically observed scaling laws across a range of modalities and architectures (Kaplan et al., 2020; Henighan et al., 2020) which show monotonic improvement of performance with increasing model scale. This view contrasts with the classical bias-variance tradeoff, which says that "once we pass a certain threshold, larger models are worse" (Nakkiran et al., 2019). The latter view is only true assuming model capacity is equal to model complexity, and would follow from bounds of the form of Equation 2. However, this appears *not* to be the case. Lotfi et al. (2024) recently established the first non-vacuous generalization bounds for large language models by producing highly compressible models trained in a non-linear low-rank subspace, and bounding their generalization performance under a smoothed entropy loss by the Kolmogorov complexity. While they focus on explicit generalization *bounds*, we focus on the complexity dynamics of the transition from memorization to generalization to explain grokking. Goldblum et al. (2023) argue for generalization to be understood using Kolmogorov complexity, and apply the Solomonoff prior, $P(h) = 2^{-K(h)}$ for hypotheses $h$ to a finite hypothesis bound (Langford & Seeger, 2001) to produce a generic generalization bound based on the Kolmogorov complexity.

**Coarse-graining**    Aaronson et al. (2014) study the complexity dynamics of a cellular automaton which simulates the mixing of two liquids. They observe the rise and fall of complexity as entropy increases, and conjecture that this phenomenon is generic. To approximate the complexity of their automaton, they propose an "apparent complexity" measure which is the compressed file size of a *coarse-grained* description of the state. They informally assert that coarse-graining removes random information from the state, leaving only "interesting" information behind. We formalize their intuition using rate–distortion theory, and use the resultant complexity measure to explain grokking.

## 3 COMPLEXITY AND GENERALIZATION

Understanding generalization is of central importance in machine learning. To what extent can we expect our models to generalize, and can we predict their performance ahead of time? Most

generalization bounds are a function of model complexity, so understanding complexity is key to understanding generalization. Lotfi et al. (2024) provide one such bound, which quantifies the expected risk $R(h)$ for a given hypothesis $h$, with probability $1 - \delta$:

$$R(h) \leq \hat{R}(h) + \sqrt{\frac{K(h) + 2 \log K(h) + \log(1/\delta)}{2n}} \tag{4}$$

Where $K(\cdot)$ is the Kolmogorov complexity, $\hat{R}(h)$ is the empirical risk, and $n$ the number of samples. As mentioned in the introduction, the Kolmogorov complexity is not computable, but can be upper-bounded by compression. Therefore, we can bound our generalization error by compressing models, with better bounds following from tighter compression. It is important to keep in mind that some approaches we might take to compress models, such as reducing the precision of their parameters, can also result in loss of performance. Hence, while we might decrease the complexity term by compressing models, this may result in the empirical risk $\hat{R}(h)$ increasing. To produce strong generalization guarantees, we must jointly minimize the model complexity and the empirical risk.

In fact, this can also be understood as a restatement of the MDL principle. If the empirical risk is understood as the entropy of the training data under the model, as is typically the case, then the joint minimization of the empirical risk and model complexity is exactly the MDL principle. I.e. the model which generalizes best is the one which compresses the data best, and is itself most compressible.

Unfortunately, naïvely compressing a model's parameters produces poor complexity bounds, since the network contains random information left over from initialization, and from the stochastic training procedure. If we could remove the random information from a network, compressing the resulting parameters would give much tighter bounds on the network complexity. However, it is undecidable whether information is random (Li & Vitányi, 1993), so there is no generic method to remove it.

Aaronson et al. (2014) remove random information by coarse-graining their automaton's state. They coarse-grain by smoothing and quantizing the state. The coarse-grained state is then compressed with off-the-shelf programs such as `bzip2`, with the compressed file size providing improved complexity estimates. They dub this metric the "apparent complexity", since it is not actually a bound on the original state's complexity, and is arbitrary in the choice and degree of coarse-graining. We adapt their method and apply it to neural networks based on a central insight: **the loss function induces a coarse-graining**, which can be understood as principled *lossy* compression of the model.

### 3.1 LOSSY COMPRESSION

Shannon's theorem provides a lower-bound on coding rates which can *losslessly* compress signals, but more efficient coding rates are possible if some distortion is allowed. For example, JPEG compression removes high-frequency details from images which are perceptually irrelevant to the human eye, incurring some distortion. Rate–distortion theory formalizes the trade-off between coding rate, induced distortion, and perceptual quality (Blau & Michaeli, 2019), though it is typically defined over random variables. Vereshchagin & Vitányi (2004) generalize the theory of rate distortion to the algorithmic information setting. They define the algorithmic rate–distortion function $r_x(y)$, which is the minimum number of bits needed to specify $y$ with distortion relative to $x$ no more than $\epsilon$:

$$r_x(y) = \min\{K(y) : d(x, y) < \epsilon\} \tag{5}$$

Where $d$ is some distortion function. We will take the "input string" $x$ to be the model parameter vector $\theta$, and the "output" $y$ to be the coarse-grained parameter vector $\hat{\theta}$. The choice of distortion function $d$ is critical. In particular, we are not interested in the distortion in parameter space, $|\theta - \hat{\theta}|^2$. Instead, we are interested in the distortion *under the loss function*. Hence, we set:

$$d(\theta, \hat{\theta}) = \left| \mathcal{L}(\theta, D) - \mathcal{L}(\hat{\theta}, D) \right| \tag{6}$$

Substituting this distortion function into Eqn 5, we can see that the optimum can be interpreted as the least complex set of model weights $\hat{\theta}$ which achieve approximately the same performance under the loss function. We control the rate–distortion trade-off through the parameter $\epsilon$, which leads to the following distortion criterion to accept a coarse-grained approximation:

$$\left| \mathcal{L}(\theta, D) - \mathcal{L}(\hat{\theta}, D) \right| < \epsilon \tag{7}$$

Note that we could have chosen an alternative distortion function, such as difference in *accuracy* or any other performance metric of interest. However, the cross-entropy loss preserves the interpretation of the model as a compressor of the training data. In fact, the MDL principle implies an optimal distortion bound $\epsilon$, which determines the best coarse-grained model $\hat{\theta}$:

$$\epsilon \leftarrow \arg\min_{\epsilon} H(D \mid \hat{\theta}) + r_\theta(\hat{\theta}) \tag{8}$$

Intuitively, as we coarse-grain a model it becomes less complex. However, it may also perform worse. The MDL principle implies that the optimally coarse-grained model is one which balances the complexity–performance trade-off, as measured by the model's ability to compress its training data, which is given by the loss function. In particular, a *lossy model* might be a better *lossless compressor* of the data, as measured by the total description length. We approximate the algorithmic rate–distortion function via compression upper bounds, and plot curves of best fit in Fig 3.

To obtain models which are optimal compressors, we must regularize them to limit their complexity. In the next section, we discuss the relationship between complexity, capacity, and regularization, and propose a regularization method which results in highly compressible models.

## 4 CAPACITY AND REGULARIZATION

To control model complexity, practitioners introduce regularizers to the network training process. Most regularizers work by limiting a model's effective capacity, which *may* limit its complexity. To see this, we consider a simplified representation of a model parameter. The information capacity $I$ of a single positive parameter with fixed precision $\delta$, and max size $\lambda$ is given by:

$$I = \log \frac{\lambda}{\delta} \tag{9}$$

Under this simplified model, it is clear that we can limit the information capacity of the parameters either by controlling their range through $\lambda$, or by reducing their precision through $\delta$, which can be thought of as an effective quantization level. Regularizers such as weight decay control the information capacity by limiting $\lambda$, or proxies of it such as an $L^p$ norm. In particular, weight decay adds an independent $L^2$ penalty for each parameter to the loss function. Methods which control information capacity through $\delta$ include training in reduced precision (Micikevicius et al., 2017), and quantization-aware training (Ma et al., 2024).

However, training with regularization does not guarantee that the resultant model will be of low complexity. For example, while weight decay ensures parameter norms are smaller than they otherwise would have been, the network can still store detail in high-frequency information. Our experiments demonstrate that this indeed occurs in practice. A key requirement for complexity reduction therefore is a regularization scheme which explicitly alleviates this issue.

A further complication is information which is spread across multiple parameters. Networks are typically parameterized by matrices which represent linear transformations, such that relevant information exists in their *spectrum*. Ultimately such information is accounted for by the total sum of parameter capacities, but it is helpful to think of information in the spectral representation.

We now introduce a regularization scheme which addresses both of these challenges.

### 4.1 QUANTIZATION AND NOISE

In contrast to methods like weight decay, which regularize the capacity of networks by controlling their range through effective penalties on $\lambda$, we introduce a simple modification to the training procedure which has previously been studied by Hinton & van Camp (1993), amongst many others: One way to set an effective lower bound on the quantization level $\delta$ from Eqn. 9 is to train networks with noisy weights. The noise prevents the parameters from forming structure below the noise threshold, limiting the information capacity "from the bottom". To do this, we construct a set of noisy weights $\tilde{\theta}$ by adding independent Gaussian noise scaled by $\delta$ to every parameter at each forward pass:

$$\tilde{\theta} = \theta + \delta\mathcal{N}(0, 1) \tag{10}$$

We compute the forward pass and gradients with the noisy weights, then apply those gradients to the original parameters. Note that $\delta$ in Eqn 10 need not be the same as $\delta$ from Eqn. 9, but applying

this training procedure sets an effective lower bound on $\delta$ from Eqn. 9 (the effective precision of our parameters) so we use the same symbol. This implies that we ought to be able to remove all information below the scale set by $\delta$, i.e. that we can quantize the parameters with bin spacing at least $\delta$. To do this, we introduce a quantization operator $\hat{Q}$ which takes a parameter vector $\theta$ and bin size $\Delta$, and rounds all parameters to the nearest bin value

$$\hat{Q}(\theta, \Delta) = \lfloor \frac{\theta}{\Delta} \rceil \times \Delta \tag{11}$$

We could achieve greater compression by using a separate $\Delta$ for each parameter group, but for simplicity we use a global value of $\Delta$. A simple algorithm which estimates the complexity of a network based on quantization coarse-graining alone is given in Algorithm 2 in the appendix.

## 4.2 SPECTRAL ENTROPY

While coarse-graining by quantization is simple and effective, it can only address high-frequency noise. In this section we introduce another coarse-graining procedure which takes advantage of low-rank approximation to remove random information. We use this low-rank approximation both as a regularization method to limit model complexity, and as a coarse-graining method during the compression stage, to produce tighter complexity estimates throughout training.

Low rank approximation is the problem of finding an approximating matrix $\tilde{M}$ for a matrix $M$ subject to the constraint that $\text{rank}(\tilde{M}) \leq r$ for some $r$, typically $r \ll \text{rank}(M)$. Low-rank approximation of neural network weight matrices has recently been used to achieve compute– and parameter–efficient finetuning of LLMs (Hu et al., 2021; Dettmers et al., 2023), building on the insight that both the weights and update gradients live on a low-dimension subspace of the total weight space (Li et al., 2018; Aghajanyan et al., 2020).

The mismatch between the apparent and effective ranks of the largest models is another manifestation of the difference between the capacity and complexity of networks. While large capacity may be needed for training dynamics to find optimal solutions, those learned solutions which generalize best tend to be of low effective rank, and therefore low complexity.

It is well-known that the optimal rank $k$ approximation of $M$ under the Frobenius norm is obtained by computing the singular value decomposition of $M$, and removing all but the $k$ largest singular values. Recall that for an $m \times n$ matrix $M \in \mathbb{R}^{m \times n}$, the singular value decomposition is $M = USV = \sum_{i=1}^{r} \sigma_i u_i v_i^\mathsf{T}$, where the columns of $U$ and the rows of $V$ are the left- and right-singular vectors respectively, and $S$ is a diagonal matrix $S = \text{diag}(\sigma_i)$ with singular values $\sigma_1 > \sigma_2 > ... > \sigma_{r=\min(m,n)}$ in decreasing order. Then the optimal rank $k$ approximation is $\tilde{M} = \sum_{i=1}^{k} \sigma_i u_i v_i^\mathsf{T}$.

To coarse-grain, we search for the smallest rank decomposition of each layer that satisfies the $\epsilon$–bound of Eqn. 7. Note that different layers may admit different degrees of rank decomposition, so using a global value of $k$ across all layers of the network would be suboptimal. Instead, we search for optimal per-layer low-rank approximations by normalizing the singular values

$$\bar{\sigma}_i = \frac{\sigma_i}{\sum_j \sigma_j} \tag{12}$$

so that $\sum_i \bar{\sigma}_i = 1$. Next we define a parameter $\tau \in (0, 1)$ which sets an approximation threshold, so that the sum of the normalized singular values is no less than $\tau$. Intuitively, $\tau$ is a parameter which allows for convenient control of the approximation quality of the low-rank decomposition. For example, a matrix with only one large singular value, and many other negligibly small singular values has an effective rank of 1, so we will probably be able to satisfy the $\epsilon$–bound with a rank 1 approximation of that matrix, whereas for a matrix with many singular values of similar magnitude, our approximation must be close to full rank. To represent each layer with as few parameters as possible, we define a per-layer truncation threshold $k(\tau)$:

$$k(\tau) = \min\{k : \sum_{i=1}^{k} \bar{\sigma}_i \geq \tau\} \tag{13}$$

where the same threshold $\tau$ is used across all weight matrices. This lets us define a low-rank approximation operator $\tilde{R}(\theta, \tau)$ which returns the truncated singular value decomposition with

effective rank controlled by $\tau$. Note that if the effective rank of the matrix is large, there is no compression benefit to the spectral representation. An $m \times n$ matrix $M$ normally requires $mn$ parameters for representation, and the SVD of rank $k$ uses $k(m + n)$ parameters, so the low-rank approximation is only useful if $k(\tau) < \frac{mn}{m+n}$.

$$\tilde{R}(\theta_j, \tau) = \begin{cases} \left(\sigma^j_{i:k(\tau)}, u^j_{i:k(\tau)}, v^j_{i:k(\tau)}\right) & \text{if } k(\tau) < \frac{mn}{m+n} \\ \theta_j & \text{otherwise} \end{cases} \tag{14}$$

Where $\theta_j$ is the network parameter group of layer $j$, and $(\sigma^j_{i:k(\tau)}, u^j_{i:k(\tau)}, v^j_{i:k(\tau)})$ are the corresponding low-rank matrices of rank $k(\tau)$. Note that we can interpret the normalized singular values as a probability distribution. Then, define the *spectral entropy* of a matrix $H_{\text{svd}}$,

$$H_{\text{svd}} = -\sum_i \bar{\sigma}_i \log \bar{\sigma}_i \tag{15}$$

Intuitively, $H_{\text{svd}}$ tells us about the effective rank of the matrix by measuring the spread of the singular values. To see that, note that if all singular values are equal, $H_{\text{svd}}$ is maximized, and if all but one of the singular values are zero, $H_{\text{svd}}$ is minimized. Roy & Vetterli (2007) formalize this by proving that $H_{\text{svd}}$ computes the effective rank of a matrix, $\text{rank}_{\text{eff}}(M) = \exp(H_{\text{svd}}(M))$, which measures the intrinsic dimension of the transformation parameterized by $M$. The spectral entropy not only provides a means of measuring the effective dimension of networks as training progresses—we can also explicitly penalize this quantity to encourage the network to learn simple, low-rank representations. In experiments, we show that our regularization method causes grokking, and produces the lowest-rank and most compressible networks of the methods we study.

Our full coarse-graining procedure is to perform a Bayesian optimization for the values of $\tau$ and $\Delta$ which produce the most compressed description, while satisfying the $\epsilon$–bound. When the low-rank description is more compressible, we quantize the *decomposed* matrices. The full procedure is given in Algorithm 1. We emphasize that the low-rank decomposition is used for two separate purposes: 1) the spectral entropy penalty is used as a regularizer, and 2) the low-rank decomposition is used to obtain compressed descriptions of the model at the coarse-graining stage.

The regularizer we study in experiments adds the spectral entropy penalty scaled by $\beta$ to the loss,

$$\mathcal{L}_{\text{reg}}(\theta) = \beta H_{\text{svd}}(\theta) \tag{16}$$

along with the noisy gradients described in section 4.1, plus weight decay via the AdamW optimizer.

---

**Algorithm 1:** Bayesian Optimization for Compression by Coarse-Graining

---

**Input:** Neural network parameters $\theta$, number of optimization steps $N$, error tolerance $\epsilon$
**Output:** Optimal compressed size
Initialize Bayesian Optimizer $BO$;
best_compressed_size $\leftarrow \infty$;
**for** $i = 1$ **to** $N$ **do**
    $\tau_i, \Delta_i \leftarrow BO.\text{SUGGESTPARAMETERS}()$;
    $\theta_R \leftarrow R(\theta, \tau_i)$ ;                         // Low-rank approximation
    $\theta_{RQ} \leftarrow Q(\theta_R, \Delta_i)$ ;                         // Quantization
    **if** $|\text{LOSS}(\theta_{RQ}) - \text{LOSS}(\theta)| < \epsilon$ **then**
        compressed_size $\leftarrow \text{COMPRESS}(\theta_{RQ})$;
        **if** *compressed_size < best_compressed_size* **then**
            best_compressed_size $\leftarrow$ compressed_size;
        **end**
        $BO.\text{UPDATEMODEL}(\tau_i, \Delta_i, \text{compressed\_size})$;
    **else**
        $BO.\text{UPDATEMODEL}(\tau_i, \Delta_i, \infty)$ ;      // Penalize invalid solutions
    **end**
**end**
**return** best_compressed_size

---

## 5 EXPERIMENTS

To study the complexity dynamics of networks transitioning from memorizing to generalizing solutions, we adopt the "grokking" tasks first reported in Power et al. (2022). These are a set of modular arithmetic tasks which networks easily over-fit, quickly achieving zero training error while test error remains large, suggesting memorization of the training examples. After many further steps of gradient descent after over-fitting, networks suddenly generalize and achieve perfect test accuracy. Although grokking is a sign of improperly regularized training, the extended delay between memorization and generalization is convenient for understanding complexity dynamics, and their relationship with generalization.

We demonstrate a regularizer, $\mathcal{L}_{\text{reg}}$ on the grokking tasks and show that 1) it causes grokking 2) it results in lower complexity networks than weight decay alone, the most widely-used regularization method, as measured by our complexity metric.

We reproduce a subset of the grokking experiments first reported in Power et al. (2022). Our training tasks all consist of learning a binary operation mod a prime number $p = 113$. For each binary operation we construct a dataset of equations of the form $\langle x \rangle \langle \text{op} \rangle \langle y \rangle \langle = \rangle \langle x \circ y \rangle$, where $\langle a \rangle$ stands for the token corresponding to element $a$. We train standard decoder-only transformers with a single layer for mod addition and multiplication, and two layers for mod subtraction and division. Further training details can be found in the appendix. We use Algorithm 1 to compute complexity upper-bounds on the coarse-grained models. The final compression step is performed by `bzip2`, an off-the-shelf compression utility.

Figure 10 in the appendix is a sanity check to ensure that grokking occurs: indeed, the regularized models generalize, while the unregularized models never achieve generalization. Fig 2 plots accuracy and complexity together for our regularized models, and indicates that as we expected from Occam's Razor, complexity is maximized during memorization, and falls during generalization. Next, we plot algorithmic rate–distortion curves of best fit in Fig 3. We fit curves of the form $y = a \log(x) + b$, and display $R^2$ values in the legend. These curves show how lossy *model* compression trades off with lossless *data* compression for the converged models at varying distortion bounds.

We compare the total description length of all models in Fig 4, and find that the models produced by our regularizer achieve the best explanation of the dataset amongst the methods. That is, they achieve a smaller sum of model complexity and data entropy vs the other methods, hence achieve the greatest compression of the data in this model class. Plotting the generalization bound from Eqn. 4, we see in Fig 5 that again, the models produced by our regularizer come closest to a non-trivial generalization bound. The goal of this work is to explain grokking, clarify the nature of complexity and its relationship to generalization, and provide a practical method of complexity approximation based on insights from lossy compression. In particular, we explained how lossy compression provides a principled approach to noise reduction and coarse-graining in networks. In particular, we did not attempt to achieve the tightest possible complexity bounds. Nevertheless, it is surprising that we get as close as we do to non-trivial generalization bounds, since the models we use are large, and the datasets small: $|D| = 12769$ vs. $|D| \approx 1.5 \times 10^{13}$ for contemporary LLMs (Dubey et al., 2024).

## 6 CONCLUSION

This work examines the complexity dynamics of neural networks, and proposes an explanation of the *grokking* phenomenon—the sudden transition from memorization to generalization long after the training data is over-fit. We do this by introducing a novel measure of intrinsic complexity based on Kolmogorov complexity and rate-distortion theory. We observe a consistent pattern of the rise and fall of complexity during training, corresponding to memorization followed by generalization. Our principled approach to lossy compression of neural networks formalizes the concept of "apparent complexity" and connects it to explicit generalization bounds. Our proposed spectral entropy regularization method encourages low-rank representations by penalizing a differentiable measure of the model's effective dimension, which helps limit model complexity and enables tight compression bounds. We hope that a deeper view of complexity beyond mere parameter count and capacity measures will lead to greater insight into the nature of generalization in learning systems.

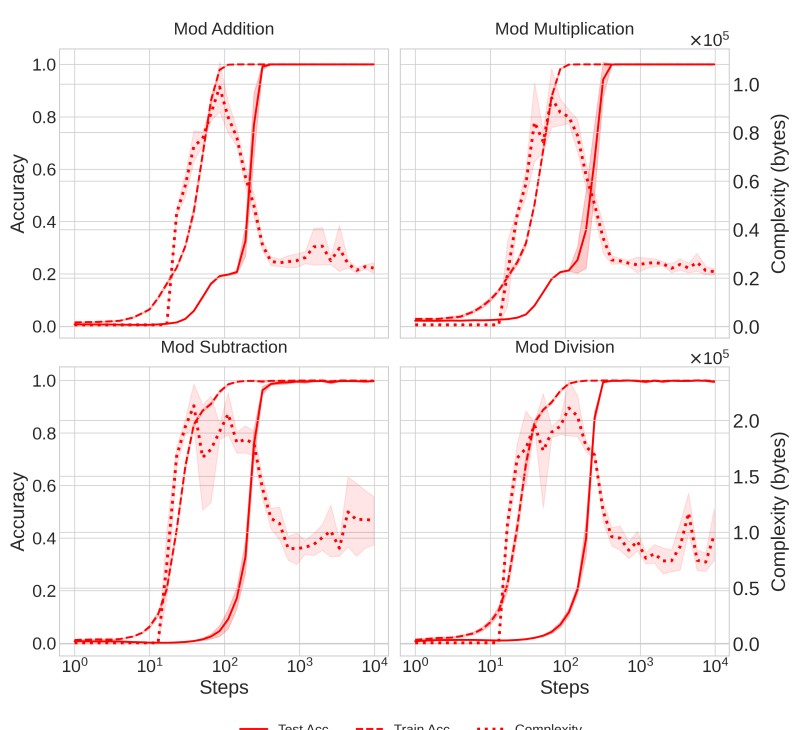

Figure 2: Complexity and accuracy vs steps for models regularized using our method. As models memorize, complexity increases. Complexity falls as generalization occurs. We use Algorithm 1 to coarse-grain models with $\epsilon = 1$. Plots show the mean of six seeds with std. error shaded.

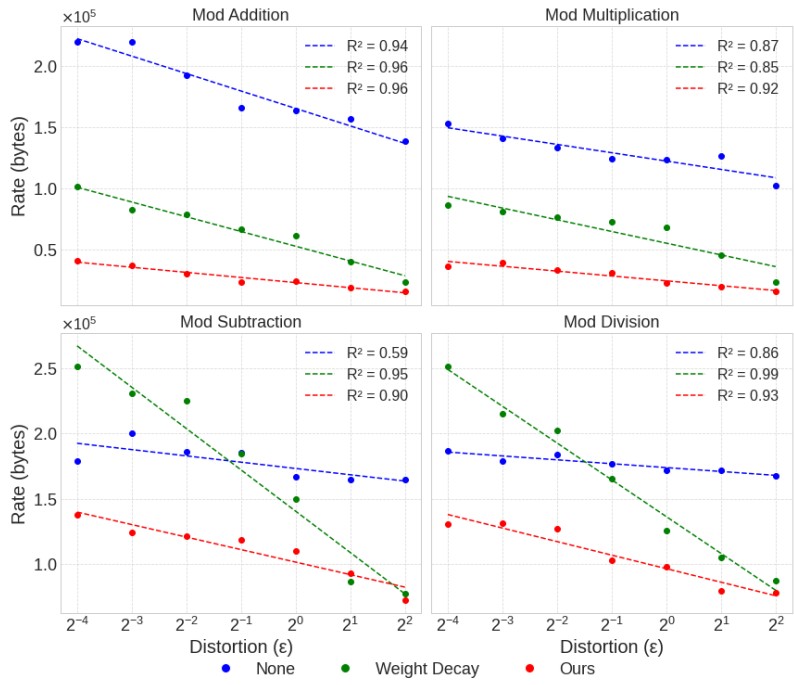

Figure 3: Algorithmic rate–distortion curves $r_\theta(\hat{\theta})$ for all datasets at varying distortion bounds $\epsilon$. The rate is the complexity of the coarse-grained weights $\hat{\theta}$ at the end of training. Mean over six seeds.

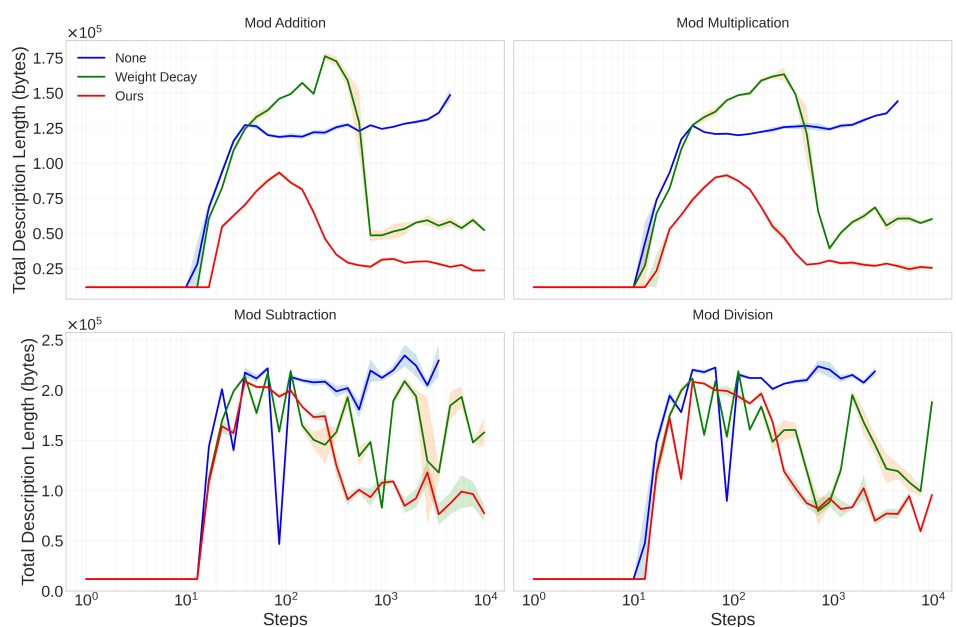

Figure 4: Total description length (complexity + entropy) vs training steps. Models trained with our regularization (red) have a smaller total description length compared to baselines. The unregularized model collapses during memorization in the subtraction and division tasks. Apparent complexity computed for $\epsilon = 1$.

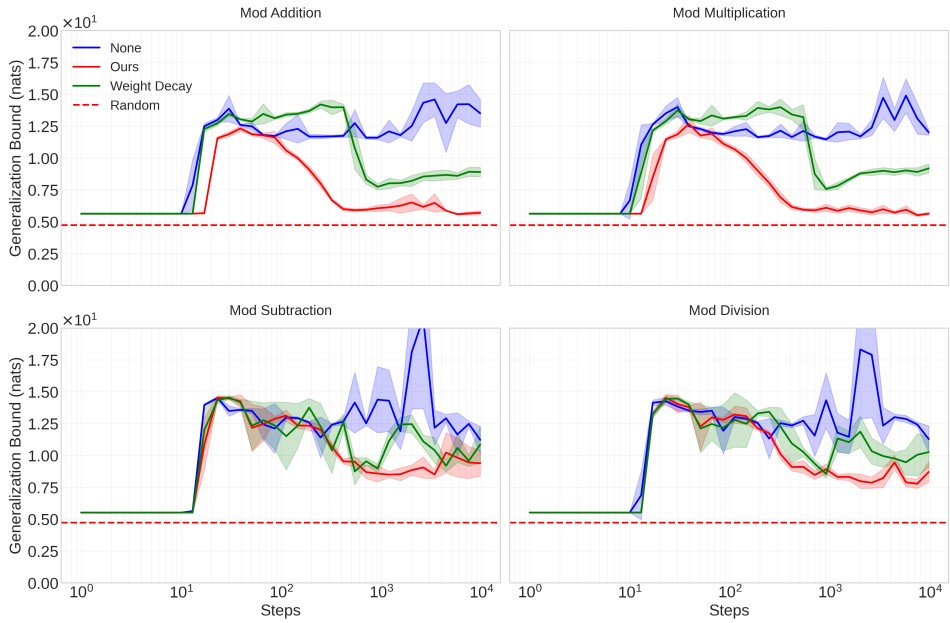

Figure 5: We plot the generalization bound on the expected risk according to Eqn. 4. While no models achieve low enough complexity to give non-vacuous bounds, our regularization method achieves the best bounds in all cases. It is not surprising that we can only achieve a vacuous bound, since the models are vastly over-parameterized, and the dataset small.

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

## A  APPENDIX

---

**Algorithm 2:** Complexity by Quantization

---

**Data:** params, epsilon
**Result:** best_complexity
best_complexity ← COMPRESS(params);
**for** *bin_sizing* **from** *small* **to** *large* **do**
    quantized_params ← QUANTIZE(params, bin_sizing);
    **if** $|\text{LOSS}(quantized\_params) - \text{LOSS}(params)| < \epsilon$ **then**
        best_complexity ← MIN(COMPRESS(quantized_params), best_complexity);
    **end**
**end**
**return** best_complexity

---

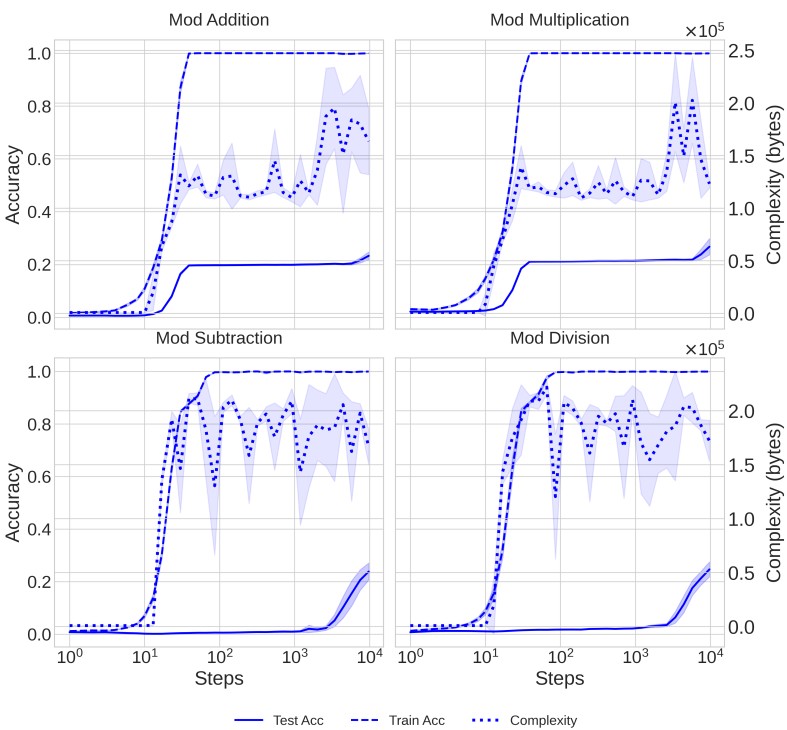

Figure 6: Complexity and accuracy vs steps for unregularized models. As models memorize, complexity increases. Without regularization, the models fail to find low complexity solutions, and never generalize. We use Algorithm 1 to coarse-grain models with $\epsilon = 1$.

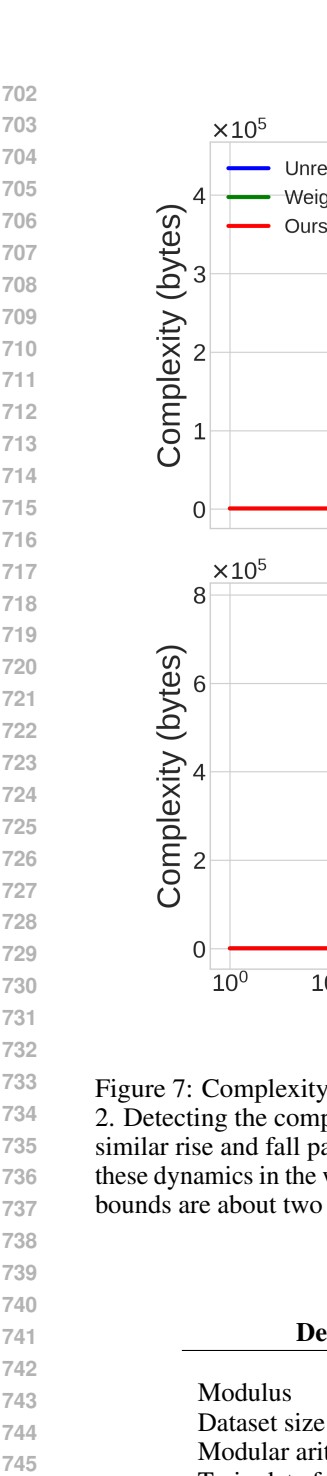

Figure 7: Complexity estimates using quantization coarse-graining alone, according to Algorithm 2. Detecting the complexity dynamics requires relatively tight complexity bounds. While we see a similar rise and fall pattern in the regularized models of the top row, this method fails to distinguish these dynamics in the weight decayed models in the subtraction and addition. Note that the complexity bounds are about two times worse than those achieved by the spectral method in Fig 2.

Table 1: Hyperparameters

| Description | Symbol | Value |
|---|---|---|
| Modulus | $p$ | 113 |
| Dataset size | $N$ | $p^2 = 12769$ |
| Modular arithmetic tasks | $\{+, \times, -, \div\}$ | - |
| Train data fraction per task | - | $(+, \times : 20\%), (-, \div : 30\%)$ |
| Transformer layers | $n_{\text{layers}}$ | $(+, \times : 1), (-, \div : 2)$ |
| Transformer hidden dimension | $d_{\text{model}}$ | 128 |
| Transformer heads dimension | $d_{\text{head}}$ | 32 |
| Transformer head count | $n_{\text{head}}$ | 4 |
| Transformer MLP dimension | $d_{\text{mlp}}$ | 512 |
| Noise regularization scale | $\delta$ | $(+, \times : 10^{-2}), (-, \div : 10^{-3})$ |
| Spectral entropy scalar | $\beta$ | $10^{-1}$ |
| Optimizer | - | AdamW |
| Weight decay scalar | - | 1 |
| Learning rate | - | $10^{-3}$ |

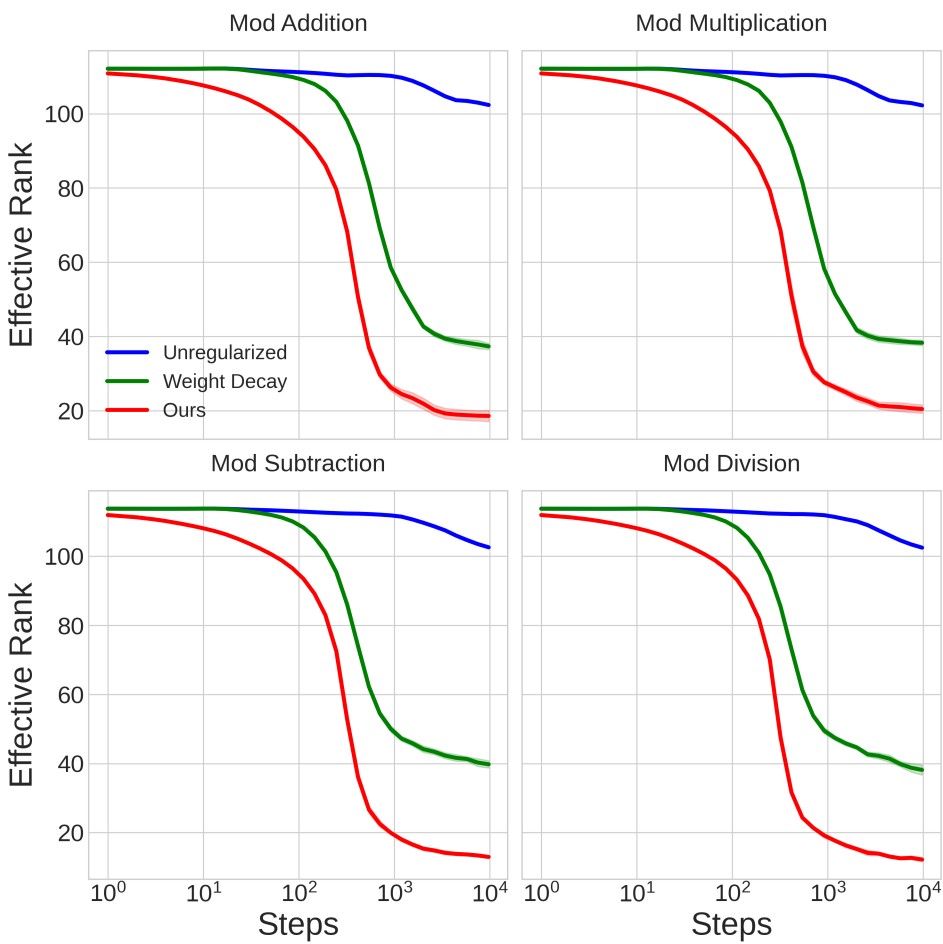

Figure 8: Effective rank ($\text{rank}_{\text{eff}} = e^{H_{\text{svd}}}$) of models throughout training. We see that the regularized models fall in effective rank when generalizing, while the unregularized model remains near full rank throughout training. Our regularizer explicitly penalizes this measure, and results in both the lowest rank and most compressible model amongst those we study.

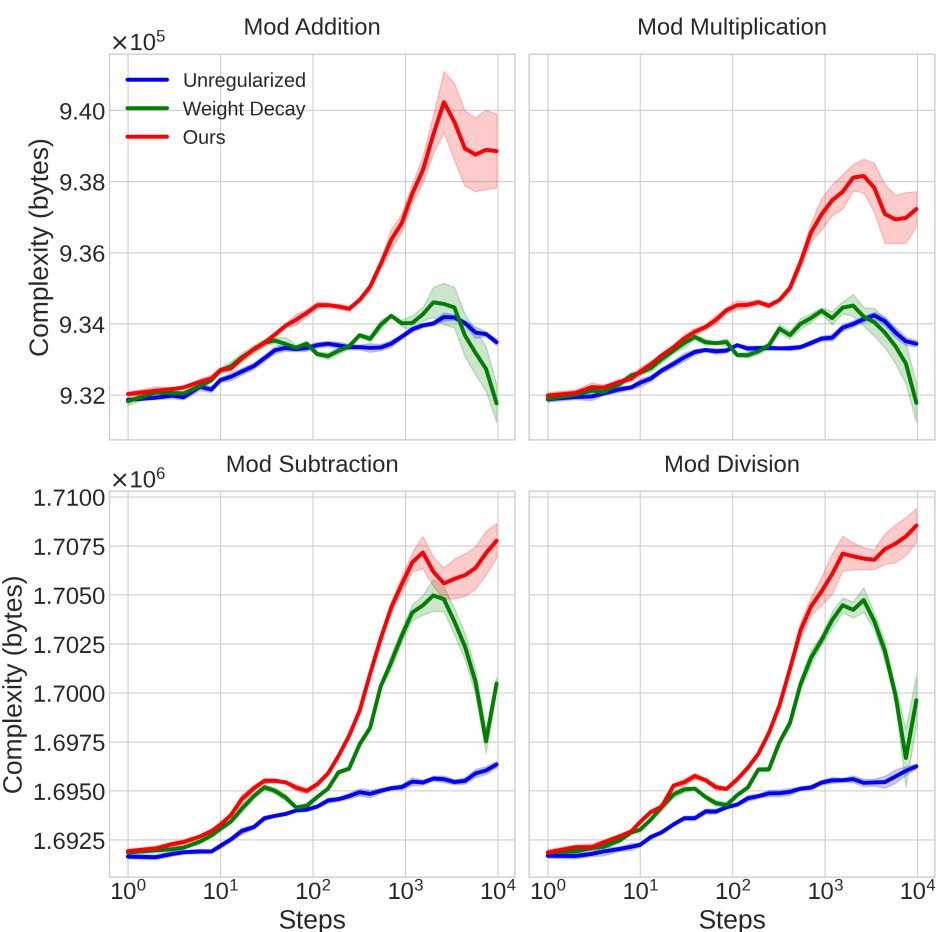

Figure 9: In this figure, we simply plot the naïve bzip2 compression of the model weights without any coarse-graining (note the y-axis scales). The range of variation in these complexity estimates is very small throughout training—if we plotted it on the same scale as other complexity plots, it would simply look like a flat line from the starting estimate. The dynamics here may correspond to weight patterns induced by the different regularizers that the compression algorithm is particularly suited or unsuited to compressing. Intriguingly, the dips in the complexity bounds of the regularized models around $10^3$ steps corresponds to generalization, giving some slight hint that some change has occurred in the model. To fully reveal this phenomenon, the coarse-graining procedure is required.

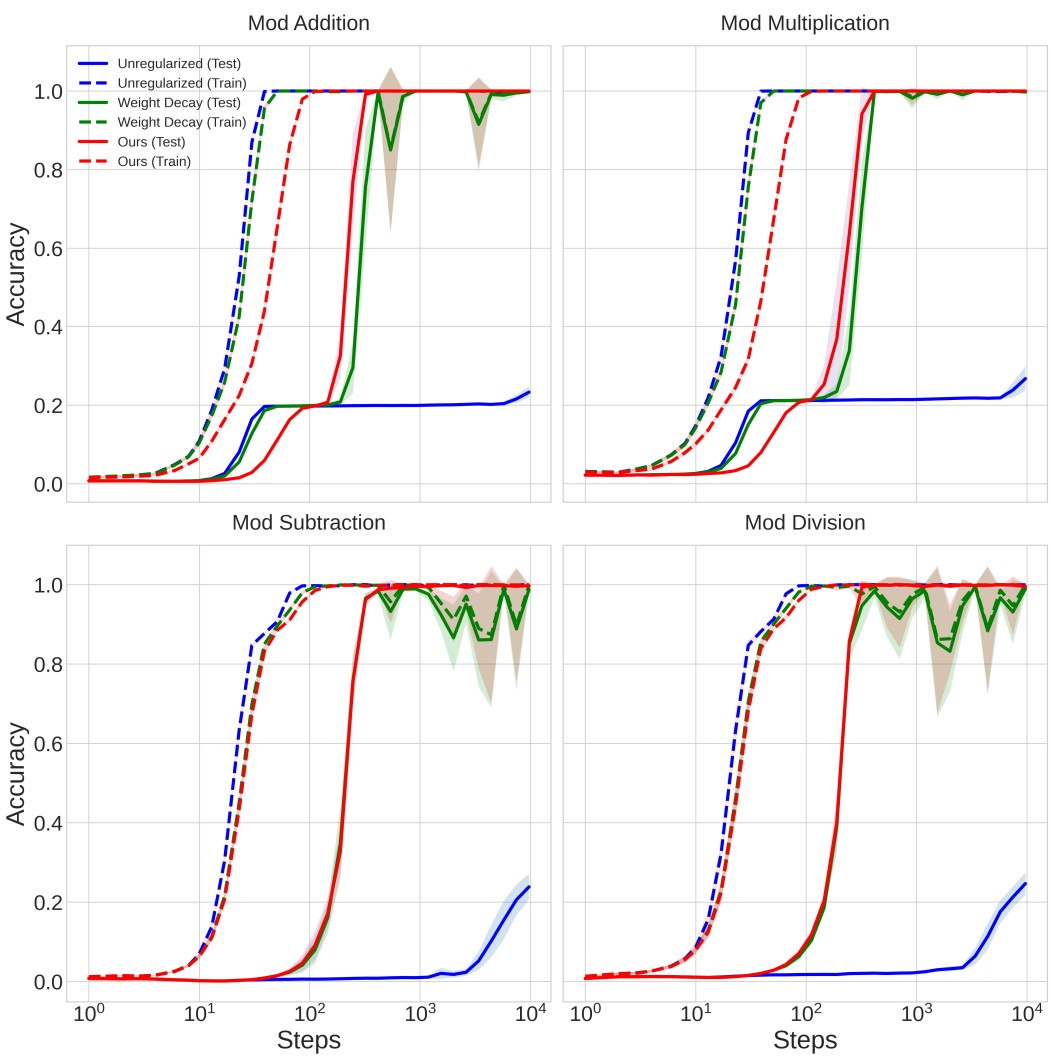

Figure 10: Test and train accuracy vs steps. Grokking occurs in all regularized models, and does not occur in unregularized models. We plot the mean over six seeds, and shade the std. error.