# OpenReview forum: "The Complexity Dynamics of Grokking"
_ICLR.cc/2025/Conference — Submitted to ICLR 2025_

### Official Review · Reviewer_JG45 · 2024-10-21

**Soundness:** 2
**Presentation:** 2
**Contribution:** 2
**Rating:** 6
**Confidence:** 3

**Summary:**

This paper studies the grokking phenomenon through compression-based approaches. Inspired by recent work on the intrinsic complexity of neural networks, and combining it with ideas from rate-distortion, quantization and low-rank approximation, the authors propose a new measure of neural networks complexity, which consists essentially of a coarse-graining procedure. They conduct experiments on simple arithmetic tasks which demonstrate that the rise and fall of this complexity might be predictive of the network starting to generalize. Moreover, this leads them to propose a new regularization scheme, based on spectral entropy, whose effect seems to reduce the total description length and the generalization bound, compared to other methods. This might lead to non-vacuous generalization bounds.

**Strengths:**

- Grokking is an important topic for the community

- The experiments suggest that the proposed regularization technique based on spectral entropy may induce grokking, which may be of practical interest.

- The experiments suggest that the rise and fall of the proposed complexity seems to be predictive of when the model starts to generalize.

- The proposed regularization techniques lead to better generalization bounds than classical weight decay or no regularization.

**Weaknesses:**

- Several notions are mentioned repeatedly but without being formally defined, such as capacity, distortion or $(\lambda,\delta)$ (Equation (9)). It would improve the paper to include additional theoretical background and more formal definitions.

 - It should be made clearer how the quantities introduced in Sections 3.1 and 4 are related to generalization. For instance, is it possible to write down a theorem with explicit dependence on these quantities, or are their consideration partially based on intuitions? Can the link of these quantities with Kolmogorov complexity be made more formal?

 - Despite the lack of formal theorems and proofs, the experiments are done on very simple arithmetic tasks. Therefore, it is not clear (neither theoretically nor empirically) whether the results may be generalized to more complex settings. I think that at least one experiment on a small dataset like MNIST or CIFAR10 could improve the paper.

 - It would be useful to include an experiment comparing the performance (in terms of accuracy) with and without the proposed regularization scheme. Indeed, we see that it reduces the MDL and the generalization bound, but, if I am correct, it is not clear whether it achieves better performance overall.

 - We see in Figure 4 that the proposed regularization scheme achieves the lowest complexity. However, the complexity is computed by Algorithm 2 and the proposed regularization is precisely penalizing the quantity computed by algorithm 2. Therefore it does not seem surprising that it is the lowest. As an ablation study, it would be interesting to make the comparison using other complexity notions. For instance, using the actual test accuracy would be very informative, to see whether the proposed regularization leads to better performance.

**Questions:**

- Is it possible to perform the same experiments on more complex but still relatively simple datasets like MNIST or CIFAR10?

 - Does the generalization bound of Equation (4) only hold for finite hypothesis spaces? If yes is that a realistic assumption in practical learning settings? Moreover, could you be more precise as to why the choice of Solomonoff prior should lead to tighter bounds than other priors, such as the uniform prior over $\mathcal{H}$?

 - Line 181: Why can the empirical risk be understood as the entropy of the data under the model? Is there a way to formalize this fact?

 - Is it possible to obtain a formal statement relating the information capacity (Equation (9)) to generalization?

 - To what size and precision do the parameters $\lambda$ and $\delta$ (Section 4) refer to in practice?

 - How would the training accuracy be affected by the addition of Gaussian noise in practical deep learning settings?

 - Can you define more precisely the notations used in Algorithm 2, such as BO.SUGGESTPARAMETERS()? More generally, can you provide more details on the Bayesian optimization procedure?

 - Does your regularization technique always lead to lower test accuracy compared to weight decay?

 - Figures 3 and 5 are not analyzed in the text, can you add some insights on the result they present?

**Remarks/questions regarding lines 152 - 155 and Equation (4)**
Even though it is not central to the paper, I have some questions about this part:
As I understand it, the bounds in terms of Kolmogorov complexity are obtained by choosing a good prior distribution in the bound of Langford and Seeger. It is not clear to me that such a choice of prior provides the most useful bound. More precisely, let $\mathcal{H}$ be a finite set of hypothesis and $\sigma : \mathcal{H} \to \mathcal{H}$ be any bijection of $H$. Then $h \mapsto 2^{K(\sigma(h))}$ may be used as a prior instead of the usual Solomonoff prior, hence leading to a generalization bound in terms of $K(\sigma (h))$. Yet another possibility would be to use the uniform prior over $\mathcal{H}$. Therefore, choice of prior, and therefore the choice of Kolmogorov complexity as a generalization measure, seems to be arbitrary (please correct me if I am mistaken). Can you provide more insights as to why this leads to the most informative bound?

I would be happy to discuss this further, please correct me if I misunderstood something.


**Other minor remarks and typos**

 - In the introduction, the terms capacity and complexity are used before being defined, which may render the introduction hard to read. In general, more formal definitiosn of these concepts might enhance the readability of the paper. It could also help to define the notion of distortion function.

 - Line 122: regulariztion $\to$ regularization

 - Equation (4): there is a missing parenthesis in $\log(1/\delta)$

 - There might be a clash of notation between the parameter $\delta$ in Equations (4), (9) and (10). It would be clearer to use a different letter in each of these equations.

---

> ### Author Response · Authors · 2024-11-15
> **Response 1.1**
>
> Thank you for your review.
>
> >Several notions are mentioned repeatedly but without being formally defined, such as capacity, distortion or (λ,δ) (Equation (9)). It would improve the paper to include additional theoretical background and more formal definitions.
>
> **Formal definitions**: We formally defined all of these. The distortion is defined in Equation 6: it is the absolute difference in loss between the original and coarse-grained weights on the data. \lambda and \delta are defined just before Equation 9. They are the max size and precision of a single parameter. Imagine a parameter which can take values as large as 100 (size), but only in multiples of 10 (precision). Then its information capacity is log(100/10) = log(10) = log(number of bins).
>
> __
> > It should be made clearer how the quantities introduced in Sections 3.1 and 4 are related to generalization. For instance, is it possible to write down a theorem with explicit dependence on these quantities, or are their consideration partially based on intuitions? Can the link of these quantities with Kolmogorov complexity be made more formal?
>
> The algorithmic rate–distortion function returns the Kolmogorov complexity of the coarse-grained model at the distortion level \epsilon, under the distortion function. This means that the generalization performance can be bounded by Equation 4, which links Kolmogorov complexity with expected risk. The dependence is explicit, not based on intuition. In the revised paper we have added a plot of the rate–distortion curve (Fig 3), so that you can see the complexity levels K at different levels of distortion \epsilon. This plot shows that our method Pareto-dominates weight decay at all distortion levels. That is, our method results in more compressible models compared to weight decay at every distortion level.
>
> __
>
> > Despite the lack of formal theorems and proofs, the experiments are done on very simple arithmetic tasks. Therefore, it is not clear (neither theoretically nor empirically) whether the results may be generalized to more complex settings. I think that at least one experiment on a small dataset like MNIST or CIFAR10 could improve the paper.
>
> As mentioned above, our bounds are explicit *and universal*, unlike other complexity proxies like L2 norm. It is not a limitation of our work that we test on “simple tasks”, it is a choice. We are trying to explain grokking, which was originally demonstrated on these modular arithmetic tasks [3]. They are an excellent test-case because they provide a clear, delayed transition from memorization to perfect generalization. This is a theory paper. Not every work benefits from scale: some ideas are best demonstrated in simple settings. While we intend to scale up our work, and have already begun follow-up work which does this, here we are focused on the basic science of learning, complexity, and generalization.
>
> __
>
> > It would be useful to include an experiment comparing the performance (in terms of accuracy) with and without the proposed regularization scheme. Indeed, we see that it reduces the MDL and the generalization bound, but, if I am correct, it is not clear whether it achieves better performance overall.
>
> This is already included in the original draft’s appendix, in the final figure. As mentioned, all regularized models grok (that is, they transition from memorization (100% train accuracy, low test accuracy) to generalization (perfect test accuracy)), and the unregularized models remain in the memorization phase. The full train and test accuracy plots are shown in the final figure, in the Appendix, which we referenced in the Experiments section. There is no performance difference whatsoever between weight decay and our method, since both generalize perfectly. This is not the point of our work. Since the train and test entropy are effectively zero, the model complexity dominates the total description length, and so our method (which drives learning towards less complex structures) achieves a better total compression of the dataset, as demonstrated in the total description length plot.
>
> __
> > We see in Figure 4 that the proposed regularization scheme achieves the lowest complexity. However, the complexity is computed by Algorithm 2 and the proposed regularization is precisely penalizing the quantity computed by algorithm 2. Therefore it does not seem surprising that it is the lowest. As an ablation study, it would be interesting to make the comparison using other complexity notions.
>
> It’s the other way around: because our complexity measure is a proper complexity metric, we want to optimize it directly. We cannot, though, since the final step is not differentiable (zipping the weights). The spectral entropy penalty encourages the network to have low effective rank, but like L2, or any other information capacity proxy, it is not a complexity metric. It is only one way the network can be complex, but there are many such ways (all possible representation spaces).

---

> > ### Comment · Reviewer_JG45 · 2024-11-19
> > **thank you for your answer - response to 1.1**
> >
> > I agree that the notions such as $\lambda$ and $\delta$ are defined in the text, but sometimes it is just defined by a term like 'max size' or `precision`. I think it would be highly beneficial to define it more formally, as the notion of precision sounds a bit vague, especially for a reader who might be new to compression-based approaches. For instance, you could include in the text the examples you provided in your answer above.
> >
> > Thank you for clarifying Figure 4 in the paper, I may have misunderstood its exact meaning in my initial review.
> >
> > Would it be interesting to perform an experiment in a setting where the generalization is not perfect? (to see whether the new regularisation improves the generalization, which might have impact in practical settings).

---

> > > ### Author Response · Authors · 2024-11-20
> > >
> > > Thanks for your responses!
> > >
> > > We will add some clarifying language around the discussion of the information capacity. Would it come across more clearly if we simply say the parameters are discrete, with max size \lambda and precision \delta? This might make it more clear how the parameters carry the complexity/information content of the model.
> > >
> > > > Would it be interesting to perform an experiment in a setting where the generalization is not perfect? (to see whether the new regularisation improves the generalization, which might have impact in practical settings).
> > >
> > > Absolutely, and we are already doing this in some followup work. Since we're both introducing a new theoretical framework of complexity, as well as explaining a well-known phenomenon (grokking) in this work, we feel the paper's clarity is best served by remaining focused on grokking.
> > >
> > > The new regularization method here is *not* our main focus. Though we use it to produce highly compressible networks, the main focus is on clarifying the relationship between complexity and generalization.

---

> ### Author Response · Authors · 2024-11-15
> **Response 1.2**
>
> > For instance, using the actual test accuracy would be very informative, to see whether the proposed regularization leads to better performance.
>
> As we mentioned above, and in the paper, all regularized nets achieve perfect test accuracy, so there is no value in comparing the test accuracy.
>
> __
>
> > Does the generalization bound of Equation (4) only hold for finite hypothesis spaces? If yes is that a realistic assumption in practical learning settings? Moreover, could you be more precise as to why the choice of Solomonoff prior should lead to tighter bounds than other priors, such as the uniform prior over H?
>
> The generalization bound of Equation 4 comes from Lotfi et al [4]. The Solomonoff prior is defined over all finite strings considered as programs. This is the most generic possible assumption that one could make for a computer model of some data.
>
> The question of why the Solomonoff prior is a good one, is very deep and interesting. There is ongoing research into the apparent simplicity bias found in nature [1, 2]. It appears to be the case that nature simply has a bias towards simpler structures, hence the Solomonoff prior is superior to a uniform prior. The ultimate nature of why this is the case is not yet clear.
>
> __
>
> > Line 181: Why can the empirical risk be understood as the entropy of the data under the model? Is there a way to formalize this fact?
>
> Lotfi et al discuss this in their work which produces the bound we use. We refer you to their work to understand the nuances of the finite hypothesis bounds. In particular, they show how to adapt entropy measures for the risk (they have to make some small changes to ensure the entropy stays bounded).
>
> In practice, one can take the risk to be the cross-entropy loss used in training. This is the intriguing link between compression and generalization: both the MDL principle and generalization bounds like Equation 4 suggest that we should take the model which minimizes the sum of data entropy and model complexity. The model which compresses the data best is the one which generalizes best. This deep fact underpins our work.
>
> __
>
> > Is it possible to obtain a formal statement relating the information capacity (Equation (9)) to generalization?
>
> The information capacity can be seen as the largest upper-bound on the model complexity. That is, the model complexity can be no greater than its information capacity. For example, a model might have 100 parameters of 10 bytes each (total 10 bytes times 100 =  1KB). However, imagine all the parameters are zero (or 1, or any constant). That model is very simple, its complexity is low. We can also imagine that each of the 100 parameters is as complex as possible (e.g. uniform random), and that there is no discernable pattern in the parameters considered as a whole: the complexity of this model is large, but it is certainly no larger than the total capacity of the model (1KB). So yes, the capacity bounds the complexity, but it is the loosest possible bound. However, one can see that an effective model compression scheme might be to try distilling a larger model into a smaller one: then if the capacity of the smaller model is low enough, we may be able to produce quite tight complexity bounds.
>
> __
>
> > To what size and precision do the parameters λ and δ (Section 4) refer to in practice?
>
> It depends on model representation specifics. E.g. models trained in float32 vs float16 will have different effective max ranges and precisions, and so on. We control the max size \lambda through weight decay, and the precision \delta through the noisy weight scheme which we discuss in section 4.1
>
> __
>
> > How would the training accuracy be affected by the addition of Gaussian noise in practical deep learning settings?
>
> There is no general answer to this question, as it can depend on the specifics of the data, model, loss function, and so on. However, adding noise to the weights is an effective regularization scheme. We cited [5] in our discussion of this scheme, and recommend it for further insight on using noisy weights as a regularization method.
>
> __
>
> > Can you define more precisely the notations used in Algorithm 2, such as BO.SUGGESTPARAMETERS()? More generally, can you provide more details on the Bayesian optimization procedure?
>
> Bayesian optimization is a generic black-box optimization procedure. Say you have input-output access to some function f(x), and you want to find its maximum value. Bayesian optimization is a generic method which receives a history of (x, f(x)) pairs, and suggests a new x each time by refining a model of the function under evaluation. In our case, we are trying to minimize the compressed size of the network, so we maximize -compressed_size. Our inputs parameters are the quantization level \Delta and the parameter \tau introduced in section 4.2 which controls the degree of rank-decomposition. We used a standard python BO, found [here](https://github.com/bayesian-optimization/BayesianOptimization).

---

> > ### Author Response · Authors · 2024-11-15
> > **Response 1.3**
> >
> > > Does your regularization technique always lead to lower test accuracy compared to weight decay?
> >
> > Both regularization methods achieve perfect test accuracy.
> >
> > __
> >
> > > Figures 3 and 5 are not analyzed in the text, can you add some insights on the result they present?
> >
> > We have updated the draft with additional discussion of the figures, and new figures. In addition to the rate–distortion curves we mentioned earlier, we also added complexity dynamics plots for the unregularized network, so you can see how the complexity stays high at all times after memorization occurs, with no generalization following. We also added effective rank plots, which show how our spectral entropy method encourages models toward low effective rank. Interestingly, weight decay also seems to encourage effective low-rank representations, though not as strongly as ours.
> >
> > Regarding your question on the permutation of hypotheses. Indeed one is free to choose any prior one wishes - if we understand your point correctly, your question amounts to whether there is a unique, canonical notion of “complexity”. Firstly, consider that the permutation (bijection) you apply to the hypotheses has itself a non-zero Kolmogorov complexity, so you're not leaving the complexity invariant by permutation. However, this leads us to the question of a unique ordering on the hypotheses/natural numbers. This is an interesting and deep question, but it is beyond the scope of our work. While there exist invariance theorems for Kolmogorov complexity, they only hold up to an arbitrary constant, so we can only make strong statements about asymptotic complexity. We recommend the textbook on Kolmogorov complexity by Li and Vitanyi if you are interested in the mathematical foundations of algorithmic complexity.
> >
> > [1] Dingle, Kamaludin, Chico Q. Camargo and Ard A. Louis. “Input–output maps are strongly biased towards simple outputs.” Nature Communications 9 (2018)
> >
> > [2] Johnston, Iain G., Kamaludin Dingle, Sam F. Greenbury, Chico Q. Camargo, Jonathan P. K. Doye, Sebastian E. Ahnert and Ard A. Louis. “Symmetry and simplicity spontaneously emerge from the algorithmic nature of evolution.” Proceedings of the National Academy of Sciences of the United States of America 119 (2021)
> >
> > [3]: Power, Alethea, Yuri Burda, Harrison Edwards, Igor Babuschkin and Vedant Misra. “Grokking: Generalization Beyond Overfitting on Small Algorithmic Datasets.” ArXiv abs/2201.02177 (2022)
> >
> > [4]: Lotfi, Sanae, Marc Finzi, Yilun Kuang, Tim G. J. Rudner, Micah Goldblum and Andrew Gordon Wilson. “Non-Vacuous Generalization Bounds for Large Language Models.” ArXiv abs/2312.17173 (2023)
> >
> > [5] Geoffrey E. Hinton and Drew van Camp. Keeping the neural networks simple by minimizing the description length of the weights. In Annual Conference Computational Learning Theory, 1993.

---

> > ### Comment · Reviewer_JG45 · 2024-11-19
> > **response to 1.2 and 1.3**
> >
> > Thank you for acknowledging that the reasons why the Solomonoff prior could be a good choice are not yet clear. I think it would be very beneficial to include such a discussion in the paper (maybe in the appendix?) as it is a basic building block of your theory.
> >
> > Regarding my question on the permutation of the hypothesis, my point is that there seems to be a lot of different way to define "Kolmogorov-inspired" complexity measures that could be used in the generalization bound (for instance the composition of Kolmogorov complexity and any permutation seems to work). Therefore, why is Kolmogorov complexity the best choice? More generally, this raises the question of whether Kolmogorov complexity is the best theoretical explanation of your experimental (compression-based) results.

---

> > > ### Author Response · Authors · 2024-11-20
> > >
> > > > Thank you for acknowledging that the reasons why the Solomonoff prior could be a good choice are not yet clear.
> > >
> > > In our previous comment on the ongoing exploration of Solomonoff priors in nature, we merely meant to point out that connecting the prior with natural phenomena is ongoing. The theoretical reasons to expect the Solomonoff prior to hold are clear, and well-established: The Solomonoff prior is the appropriate universal prior that encodes a simplicity bias.
> > >
> > > If one expects that simpler objects are more likely to occur than complex objects (where complexity is the minimum description length on a Turing machine), the Solomonoff prior exactly captures this notion, and is well-studied in Algorithmic Information Theory. In fact, the Solomonoff prior was used by Hutter [1] to construct a theoretically perfect (but uncomputable) AI agent.
> > >
> > > We will include an expanded discussion of the Solomonoff prior, and concepts from algorithmic information theory more generally, in the Appendix. Thank you for this suggestion.
> > >
> > > > Regarding my question on the permutation of the hypothesis, my point is that there seems to be a lot of different way to define "Kolmogorov-inspired" complexity measures that could be used in the generalization bound (for instance the composition of Kolmogorov complexity and any permutation seems to work).
> > >
> > > This is a subtlety of Kolmogorov complexity which is a bit tricky to explain (the results of this discussion will go in the Appendix, so please let us know if this explanation clarifies the issue for you).
> > >
> > > First, we should more precisely state that we are bounding the K complexity of these parameters *relative to a given computing environment*, that is, K(\theta) in our paper should more precisely be stated as K(\theta | Python). That being said, because any Turing-complete language (e.g. Python, C, etc...) can simulate any other Turing-machine, with only a *finite, constant* cost to switch between languages (the length of the interpreter from one language to another), the Kolmogorov complexity is, in fact, universal, up to said fixed constant (the Kolmogorov Complexity Invariance Theorem). Does this clarify the issue? [These slides](https://users.cs.duke.edu/~reif/courses/complectures/Li/KC-Lecture1.pdf) provide a decent introduction into K complexity.
> > >
> > > The compression provides an upper-bound on the Kolmogorov complexity, which we can straightforwardly connect to generalization through the bound. This gives a connection between the compressed size of the model, and a generalization bound, which is what we want. We think this understanding that the compressibility of the model controlling the generalization performance is under-studied, and here we are showing how adopting that perspective can explain what is going on with grokking: by looking at how compressible the model is at each step of training, we get a very precise picture of its complexity, which corresponds exactly with its generalization performance!
> > >
> > > The permutation you're suggesting can add either very little complexity (null permutation), or can add maximal complexity (consider a permutation map \sigma which has no description shorter than writing the entire permutation out), effectively a "random" permutation. Consider the shortest description of your \sigma as a program -- composing the network weights with \sigma adds K(\sigma) complexity to the permuted weights. Does that help clarify the issue?
> > >
> > > [1]: Hutter, Marcus. “A Theory of Universal Artificial Intelligence based on Algorithmic Complexity.” ArXiv cs.AI/0004001 (2000)

---

> > > > ### Comment · Reviewer_JG45 · 2024-11-25
> > > > **Thank you for your answer**
> > > >
> > > > Thank you for clarifying the discussion around information capacity, I think it can help a lot. In particular, I would like the term "precision" to be defined formally.
> > > >
> > > > I also thank the authors for clarifying some points related to Kolmogorov complexity and Solomonoff prior.
> > > >
> > > > I still have a few questions regarding this aspect:
> > > >  - 1. You mention that Kolmogorov complexity is universal up to a constant, but I guess the prior is then normalised for the prior to be a probability distribution, so the constant does not really matter right? (which can be a good thing actually)
> > > >  - 2. It is clearly not central to the paper but I think I misexplained my permutation argument: I didn't mean to consider the complexity associated to a permutation, but rather that if $\sigma: \mathcal{H} \longrightarrow \mathcal{H}$ is any permutation of the hypothesis set $\mathcal{H}$ (ie, the prior is a distribution on $\mathcal{H}$), then $K(h)$ can be replaced by $K(\sigma(h))$ in Equation (4) (if I am correct).
> > > >
> > > > I would like to thank the authors for taking time to clarify several of my concerns. While I believe that some work remains to be done (especially regarding clarity and formal definitions of the various introduced quantities), I will increase my score to 6 (marginally above the threshold).

---

> > > > > ### Author Response · Authors · 2024-11-25
> > > > >
> > > > > **RE: The permutation question:** When we invoke the Kolmogorov complexity, we are implicitly choosing a reference Universal Turing Machine (UTM), i.e. a computing environment/programming language. This in some sense is equivalent to "choosing a prior" for doing Bayesian probability. So if you replace $h$ by $\sigma(h)$, you **must** keep track of how the permutation changes the complexity.
> > > > >
> > > > > **RE: The normalization of the Solomonoff Prior**: Does our answer above also help to clarify this point? As we mentioned, when considering finite strings like we do in this work, one should be clear about the reference UTM, as it affects the effective "prior".
> > > > >
> > > > > **RE: Precision**: The precision is, as we mentioned, the "effective quantization" level. If we want to be completely mathematical about it: we are defining our parameters over $\mathbb{R}$, and a precision map for fixed-point arithmetic with $n$ bits after the decimal point defines an equivalence class from $\mathbb{R} \rightarrow \mathbb{Z}/(2^n)\mathbb{Z}$. I.e. the precision is a quotient set where numbers differing by less than $2^{-n}$ are considered equivalent. We consider this formalism to be cumbersome for the clarity of our paper, and prefer to leave the wording as "precision".

---

### Official Review · Reviewer_PJNM · 2024-10-26

**Soundness:** 3
**Presentation:** 3
**Contribution:** 2
**Rating:** 5
**Confidence:** 4

**Summary:**

This paper proposes to study the grokking dynamics via the lens of information theory (minimum description length). In particular, they proposed: (1) a new compression algorithm to compress the neural network; (2) a new regularizer based on spectral entropy. They show that the spectral entropy regularizer outperforms the standard weight decay to the extent that a model with lower complexity is obtained. They claimed a factor of 30-40x improvement of the compression ratio over bzip2, which is impressive (although I can't find the file size data). However, none of the compression methods achieve a non-vacuous bound, since models are vastly over-parametrized.

**Strengths:**

* The paper is well-written and very readable
* The paper presents "new" theoretical tools to analyze neural networks
* The analysis is a new angle to understand grokking

**Weaknesses:**

* This paper deals with too many things simultaneously, which makes me a bit lost. What's *the* motivation of this paper? Otherwise, the paper reads like a collection of ok-ish results but none of them is impressive enough. For example, the idea of grokking as compression has been explored by [Liu et al.], [Humayun et al.] and [Deletang et al.]. The idea of using spectral entropy as a measure is explored in [Liu2 et al.], although it is novel to regularize the network with spectral entropy (which is unfortunately expensive).
* The papers claim a 30-40x improvement in compression ratio, but I did not find and details or data.
* Although this is a more theoretical paper than an experimental paper, I am not sure about its practical implications.

**References**

[Liu et al] Grokking as Compression: A Nonlinear Complexity Perspective, arXiv: 2310.05918

[Delétang et al.] Language Modeling Is Compression, ICLR 2024

[Humayun et al] Deep Networks Always Grok and Here is Why, arXiv: 2402.15555

[Liu2 et al] Towards Understanding Grokking: An Effective Theory of Representation Learning, NeurIPS 2022

**Questions:**

* What's the key motivation of this paper?
* Could you elaborate on the comparison with bzip2? What is being compressed, problem setup, compressed file size, etc.?
* What practical implications does this paper have? I would consider a method practically useful if: (1) it can speed up grokking and/or (2) it can compress real-world datasets better than baselines.

---

> ### Author Response · Authors · 2024-11-15
> **Response 1.1**
>
> Thanks for your review.
>
> **Key Motivation**: The motivation is to understand the nature of complexity and generalization better. Why and when do neural networks generalize? How can we know if they’ve learned a good explanation? What kinds of abstractions emerge in the networks? Do they learn complicated explanations, or simple ones?
> The grokking phenomenon is the clearest example we know of where during training the networks undergo a clear phase transition between memorization and generalization, so it is a perfect test-case to understand the relationship between generalization and complexity. Our results demonstrate that networks are highly complex when they have simply memorized their training data, and that when they generalize, they become much simpler. Our complexity dynamics plots demonstrate this transition from high to low complexity clearly.
>
> **Comparison with previous work**: As discussed in the related work section, Liu et al produce a complexity proxy, but their measure cannot be used to construct a generalization bound. They do not prove that their measure guarantees generalization behavior. The same is true of Humayun et al. In contrast, our method bounds the Kolmogorov complexity, which results in an explicit generalization bound in turn. Furthermore, our construction is universal, and can be applied to any kind of parameterized model.
>
> RE: Delétang et al, this work shares our view of sequence modeling as compression, but does not study grokking. Their work is focused on dataset compression with LLMs, but not on the model complexity. Our work demonstrates that model complexity must be considered jointly with data compression to understand generalization.
>
> __
>
> > Could you elaborate on the comparison with bzip2? What is being compressed, problem setup, compressed file size, etc.?
>
> **Relationship to zip**: It turns out that all else being equal, more compressible models provably generalize better, so there is a deep connection between how compressible a model is and how well it generalizes. Ultimately, one can bound the complexity of the network by its information content (e.g. its filesize). Hence, we want to know: how much can we compress this model? If we simply try to zip the weights, we don’t achieve any meaningful compression because of the random information in the network, so we don’t get insight into the model complexity. In this work, we presented a way to get rid of the noise in the weights, and we made this procedure formal using rate–distortion theory, which is the same theory that underlies related compression schemes, such as JPEG. So in our work, we give a formal theory of network compression, and connect that theory to equations which tell us how well a network will perform on unseen examples (its generalization performance). Our compression scheme has multiple steps to get rid of noise (quantization, low rank approximation), and the final step is to zip the de-noised weights, to get a complexity measure in bytes. A proper complexity measure like ours allows us to give an explicit generalization bound which is universal.
>
> __
>
> > The papers claim a 30-40x improvement in compression ratio, but I did not find and details or data.
>
> We have added the naive bzip2 filesizes of the networks, shown in Figure 9 in the appendix. Note the y-axis scale in comparison to Figs 1 and 2. The final complexity of the regularized networks as measured by our method is 30-40x smaller than with naive bzip2.

---

> ### Author Response · Authors · 2024-11-15
> **Response 1.2**
>
> > What practical implications does this paper have? I would consider a method practically useful if: (1) it can speed up grokking and/or (2) it can compress real-world datasets better than baselines.
>
> This is a **theory paper**, submitted to the learning theory track. We are interested in the basic science of learning, complexity, and generalization. Our primary concern is to understand the nature of complexity and generalization more precisely. To this end, our method provides a universal, computable complexity measure that can be used to study complexity in any parameterized model. Furthermore, our theory connects complexity with information capacity in a fundamental way, through both quantization and low-rank decomposition. Questions of key importance to ML practitioners include: How much can we quantize our model (e.g. an LLM)? How much can a model be distilled into a smaller one (e.g. what is the lowest-rank model which can achieve this performance)? How results can help answer these questions. Even more intriguingly, in our view, is the question of emergence. What kinds of abstractions/representations emerge during learning? Are they simple or complex? What does complex even mean? Will what my model learned generalize to new examples? These are the sorts of questions that our work asks, and contributes to answering.
>
> In terms of practical results, our method causes grokking to happen faster than with weight decay alone (see final figure in the appendix). Because the models it produces are less complex than the alternatives, it also achieves better compression on its datasets, since, as we discussed, the model size must be considered as part of the total compressed size of the dataset.
>
> __
>
> > the paper reads like a collection of ok-ish results but none of them is impressive enough.
>
> No one has yet explained grokking in a way the community accepts. We think that understanding the grokking phenomenon in terms of the network complexity dynamics sheds light on the nature of abstraction formation in neural networks, and which has the potential to fundamentally change how the community understands the dynamics of complexity in learning models.

---

### Official Review · Reviewer_7uBz · 2024-10-30

**Soundness:** 2
**Presentation:** 2
**Contribution:** 2
**Rating:** 3
**Confidence:** 4

**Summary:**

The authors introduce a measure of neural networks’ complexity, and show that grokking could be explained by the rise and fall of the model’s complexity. The authors also propose methods for compressing neural networks via quantization and spectral entropy-based regularization, and empirically demonstrate their performances with modular arithmetic tasks.

**Strengths:**

- The paper is generally clear, and easy to read and interpret.
- The paper provides nice intuitions on building generalizable neural networks, especially from the model complexity perspective.
- The paper considers an interesting set of techniques for model compression with minimal performance loss, and tests them with experiments.

**Weaknesses:**

While the paper considers several promising ideas for model compression, there are a few limitations:
- While the complexity explanation of grokking is interesting, it seems to overlap with the circuit efficiency explanation proposed by Varma et al. (2023). Although the authors acknowledge that model complexity is not exactly identical to efficiency or parameter norms, the added insights in this area feel somewhat limited.
- The proposed model compression methods are quite similar to existing techniques on quantization and low-rank approximations, which raises questions about the novelty of the approach. Spectral entropy-based regularization is an interesting idea, but concerns about potential computational overhead and their applicability in more complex settings remain.
- Lastly, the applicability of entropy regularization techniques in more complex problems beyond the modular arithmetic task raises some concerns. Additional evidence or analysis demonstrating how this technique can advance the complexity-performance Pareto frontier in more difficult tasks will strengthen the paper.

**Questions:**

1. How did you set the learning rates for experiments? Does the performance of entropy regularization vary with different learning rates?
2. While entropy regularization surely helps in compressing the model, I expect that both the usual L2 regularization and the entropy regularization will achieve perfect test accuracy. Could you think of a scenario where the proposed regularization technique offers a clear performance advantage over L2 regularization?
3. Will entropy regularization also help in training larger models with more complicated datasets, where they often do not have simple representations as one-dimensional numbers?
4. Could the computational overhead of low-rank optimization become significant, especially when applied to large models? If so, how could we mitigate them?

---

> ### Author Response · Authors · 2024-11-15
> **Response 1.1**
>
> > While the complexity explanation of grokking is interesting, it seems to overlap with the circuit efficiency explanation proposed by Varma et al. (2023). Although the authors acknowledge that model complexity is not exactly identical to efficiency or parameter norms, the added insights in this area feel somewhat limited.
>
> **Formal Complexity Measure**: We strongly disagree that our added insights are limited. We were indeed inspired by Varma et al to study network complexity. The issue is that the L2 norm, which they appeal to to explain grokking, is not a complexity measure. This is a widespread misunderstanding in the ML community, which has led to much confusion. We aim to clarify that confusion with this work. Aaronson et al connect effective complexity with coarse-graining, but give no formal justification for this connection. We have formalized Aaronson et al’s insight using algorithmic rate–distortion theory, and then demonstrated the effectiveness of our proper universal complexity measure by applying it to explain grokking in terms of the fundamental information content inside a network. We believe our work has far-reaching implications for understanding generalization, model compression, and quantization schemes. Understanding the nature of generalization and its relationship to complexity is a question of **central importance** in machine learning.
>
> __
>
> > The proposed model compression methods are quite similar to existing techniques on quantization and low-rank approximations, which raises questions about the novelty of the approach. Spectral entropy-based regularization is an interesting idea, but concerns about potential computational overhead and their applicability in more complex settings remain.
>
> **Relationship to prior work**: It is true that others have studied quantization and low-rank approximation in machine learning. These are topics of fundamental importance, and we make no claim whatsoever about our use of these elements being novel per se. Our contribution is to connect these fundamental ideas with another: complexity. Why can models be quantized to different degrees? Why can they sometimes be distilled into smaller networks? Our work takes a key step toward clarifying these questions by illuminating the relationship of these basic ideas with complexity and generalization. **Can you be more specific about what concerns you have about the spectral regularization method being applied in “more complex settings”?**
>
> **Followup work**: Now that we have this complexity measure, we are producing follow-up work applying these insights to other domains and scaling our method up. We’re excited to share these results with the community, but this first paper laying out the conceptual framework and demonstrating it on the grokking tasks is, in our view, the appropriate first step.
>
> __
>
> > Lastly, the applicability of entropy regularization techniques in more complex problems beyond the modular arithmetic task raises some concerns. Additional evidence or analysis demonstrating how this technique can advance the complexity-performance Pareto frontier in more difficult tasks will strengthen the paper.
>
> **Pareto frontier**: Because we are introducing a lot of conceptually new pieces in this work, we want to stay completely focused on the clearest possible example of complexity dynamics and generalization: grokking. However, we absolutely agree regarding the question of a Pareto frontier. We have updated the paper to include a plot (Fig 3) showing how the complexity of the models differ at different distortion levels, and find that our method represents a Pareto improvement over weight decay.
>
> __
>
> > How did you set the learning rates for experiments? Does the performance of entropy regularization vary with different learning rates?
>
> As mentioned in the experiments section of the paper, we use exactly the same settings as the original grokking work, to avoid any additional complication or changes. Because of this, we only used the default rate of 1e-3. We have added a hyperparameter table to the appendix.

---

> > ### Author Response · Authors · 2024-11-15
> > **Response 1.2**
> >
> > > While entropy regularization surely helps in compressing the model, I expect that both the usual L2 regularization and the entropy regularization will achieve perfect test accuracy. Could you think of a scenario where the proposed regularization technique offers a clear performance advantage over L2 regularization?
> >
> > Yes, one can achieve perfect test accuracy with almost any regularization method. It is not difficult to get these models to generalize. The point of this work is not to propose a new regularization scheme which performs better than weight decay across a range of tasks: the point is to clarify the relationship between complexity and generalization in neural networks. As we discuss in the paper, L2 is not a valid complexity metric since networks can be arbitrarily rescaled. A proper complexity metric must have units of information. Using our complexity metric, one can construct explicit generalization bounds (Equation 4), unlike prior works which study complexity in grokking.
> >
> > __
> >
> > > Will entropy regularization also help in training larger models with more complicated datasets, where they often do not have simple representations as one-dimensional numbers?
> >
> > Yes, the spectral entropy regularization will always penalize models towards low-rank solutions. Of course, if the regularization strength is too large, this can lead to model collapse, just like any other regularization method. There is no particular relationship between the fact that grokking occurs on modular arithmetic equations, and the complexity of the models.
> >
> > __
> >
> > > Could the computational overhead of low-rank optimization become significant, especially when applied to large models? If so, how could we mitigate them?
> >
> > Ultimately, in this work we want to track the complexity as closely as possible to get a sharp picture of the complexity dynamics throughout training, to illustrate the phase transition from memorization to generalization. In real-world applications, one probably does not need to get complexity estimates this densely, and if one is only interested in the final performance of the model, they could get a complexity estimate once at the end of training.

---

> > > ### Comment · Reviewer_7uBz · 2024-11-24
> > >
> > > Thank you so much for clarifying the paper! I completely agree that understanding generalization and its relationship to complexity is a crucial question in machine learning. The finding that the memorization-to-generalization transition aligns with a drop in complexity is encouraging, as it verifies our understanding of overfitting within the community.
> > >
> > > That said, I feel this explanation might not fully capture the essence of "grokking." The rise and fall of complexity seem to almost directly follow from how memorization and generalization are defined. This leaves some critical questions still open, such as:
> > > (a) When does grokking occur?
> > > (b) How do hyperparameters like learning rate and dataset size influence the memorization-generalization transition?
> > > (c) What are the competing mechanisms that drive a network toward either memorization or generalization?
> > >
> > > I would love to hear your thoughts on these aspects as well.

---

> ### Author Response · Authors · 2024-11-25
>
> Thanks for your response.
>
> > This leaves some critical questions still open, such as: [...]
>
> These questions are what we would call "phenomenological" ones, and have been explored in [1]. The answers to these questions will depend on the specific setup such as model architecture, dataset, optimizer, and so on. The theory of *why* generalization occurs is more fundamental than the phenomenology, and as we have demonstrated in this work, is directly related to complexity. If one knows the complexity, one can produce generalization predictions *without access to a test set*, using the bounds we provide alone.
>
> > The rise and fall of complexity seem to almost directly follow from how memorization and generalization are defined.
>
> What makes you say that? No one has yet demonstrated the rise and fall of complexity in these models, because a universal complexity measure has so far been unavailable. While we agree that it is intuitive, having a formal theory is incredibly important for making progress. For example, consider the "Value Equivalence Principle" [2] in model based reinforcement learning. It states that an optimal "world model" for decision making is in fact *incomplete*, and only models the world up to variations that influence the value function. We do not want to model details of the environment which will never affect our decision making. Using the lossy compression framework we have developed in this work, one can understand the Value Equivalence Principle as exactly our algorithmic rate--distortion theory, which uses the *value function* as the distortion function. Furthermore, because of the connection we make between generalization and complexity, this provides another formal justification for why agents acting under the VEP might generalize *better* than agents which model the world in full detail. Simpler explanations generalize better!
>
> We would like to point out that this is the **learning theory** track of the conference. We have demonstrated a new approach to understanding model generalization through a **universal** complexity measure which is based on lossy compression. We hope you agree that the community will be interested in this perspective. If not, could you please explain why not?
>
>
> [1]: Liu et al. Towards Understanding Grokking: An Effective Theory of Representation Learning. NeurIPS 2022.
>
> [2]: Grimm et al. The Value Equivalence Principle for Model-Based Reinforcement Learning. NeurIPS 2020.

---

### Official Review · Reviewer_n5VB · 2024-10-31

**Soundness:** 3
**Presentation:** 4
**Contribution:** 4
**Rating:** 8
**Confidence:** 4

**Summary:**

This paper studies the phenomenon of grokking through the lens of complexity theory and rate distortion theory. It proposes ways to compress model weights:
-- Via a parameter quantization operation, as a twist on ideas of Hinton and Van Camp
-- Via a low-rank approximation operation.
The idea is compress the models up to certain rate distortion thresholds, quantified by the loss.
They find that this compression is substantially more powerful than traditional compression methods (bzip) and argue that this is a better approximation of the Kolmogorov's complexity of the model.
Using this metric, the authors perform experiments on arithmetic operations and find that the grokking phase is associated with a drop from the complexity peak. Following this idea, they propose a new regularizer that apparently increases the grokking effect.

Overall, this is a very well-written paper that lays out super interesting ideas and presents a compelling thesis and nice experiments. I am not sold on the idea that this is an explanation of grokking, but the observations and the conclusions are overall very interesting and I think this is a valuable contribution to understanding better what happens with grokking and is quite promising to improve learning performance of models.

**Strengths:**

Excellent writing, compelling ideas, nice experiments, convincing thesis, possible follow-ups.

**Weaknesses:**

Is it really an explanation of grokking or more some interesting and attractive observations?
The experiments with the regularizer are not many.

**Questions:**

Have you tried applying these ideas to more complex datasets, does it compare favorably vs weight decay techniques ?

Bzip is not ideal to compress weights... are there other points of comparisons available?

What is the efficiency of your compression method? How long does it take to compress?

---

> ### Author Response · Authors · 2024-11-15
>
> Thank you for your review.
>
> **Chosen Datasets**: We intend to study more complex datasets in our next work, but we have kept this paper focused on the original grokking tasks (without making any changes), to maintain clarity and comparability with the key prior work.
>
>
> **Discussion of What Constitutes an Explanation**: It’s an interesting philosophical question whether this constitutes an “explanation” of grokking, as you mention! What might one mean by an explanation? If we use different networks, with different activation functions, different topologies, different regularizers, the learned representations will, in all likelihood, be different. That is, the microscopic dynamics/representation of the network weights might not be the appropriate level of description, and so might not constitute an “explanation”. The point here is that some abstraction emerges in the network which lets it generalize. To use an analogy: the dynamics of both water and air are governed by the hydrodynamic equations (they follow the same emergent macroscopic phenomenon), but their microscopic components are completely different (H20 vs a mix of gases). What constitutes an “explanation” of their behavior? Does one want to appeal to the microscopic dynamics (quantum mechanics), or is it enough to show that the same abstraction emerges? Our argument in this work is that we can actually ignore the microscopic weight structures, and *explicitly* bound the generalization performance (which is what one generally cares about in ML) using the complexity, which is properly thought of as a macroscopic phenomenon in the context of coarse-graining. Ultimately, it will be up to the community to decide what constitutes an explanation for grokking.
>
> **RE: Bzip, and compression time**: Yes, it’s true these are not ideal. One could imagine some specialized compression scheme for weight matrices performing better here. However, the point of this work was not to produce the tightest possible complexity estimates at all costs, but to develop the conceptual framework and give a simple implementation. For our purposes, a simple off-the-shelf compressor like bzip2 is sufficient. We also experimented with gzip for the final compression step, which produced similar results with slightly worse compression ratios. When understanding the exact generalization performance is necessary for critical systems, one would want to use the best possible compression scheme within one’s compute budget.
>
> **Followup Work**: Now that we have this complexity measure, we are producing follow-up work applying these insights to other domains and scaling our method up. We’re excited to share these results with the community, but this first paper laying out the conceptual framework and demonstrating it on the grokking tasks is, in our view, the appropriate first step.
>
> **Additions to updated draft**: We have added a number of plots to the updated version of the paper. In particular we would like to point out the plots of the rate–distortion curves for the different regularization methods. You can see that our method Pareto-dominates weight decay at every distortion level, indicating the strong performance of our regularizer. Furthermore, we’ve added complexity plots for the unregularized network, so that you can see the complexity dynamics in the case where generalization does not occur. Finally, we also added plots showing the effective rank of the networks with different regularizers, which demonstrates how our spectral entropy regularizer enforces an effective low-rank penalty, helping us outperform weight decay in complexity.

---

> > ### Comment · Reviewer_n5VB · 2024-11-24
> >
> > Thanks for the comments, in particular in terms of what you call an explanation. I remain enthusiastic about this paper.

---

### Official Review · Reviewer_pi66 · 2024-11-03

**Soundness:** 2
**Presentation:** 2
**Contribution:** 2
**Rating:** 3
**Confidence:** 3

**Summary:**

The authors introduce a new complexity measure for neural networks and claim that this complexity measure can be used to explain 'grokking'. "Grokking" in machine learning is this idea that neural networks suddenly transition from memorization to generalization long after overfitting the training data. They show that their complexity measure correlates with this 'grokking' and then show how this complexity measure can be used to define a new regularization method which encourages low-rank representations. This regularizer is defined using the spectral entropy of the network weights.

**Strengths:**

Understanding the role of model complexity and how it should be measured is an important question in machine learning. This paper takes a good step in this direction and presents a compelling case for a complexity measure which is defined using the minimum description length and ideas from compression and information theory. The paper contributes to a deeper understanding of this 'grokking' phenomenon, which has gotten significant attention in recent years.

The paper has good theoretical motivation and makes an interesting connection with the concept of grokking in machine learning. Their intrinsic complexity measure and regularization technique are well-grounded in theoretical concepts from information theory. The authors provide clear explanations and justifications for their design choices.

The paper is logically structured and well-written and supports their theoretical claims with experiments on synthetic tasks, like modular arithmetic, for decoder transformer models.

**Weaknesses:**

The complexity measure defined and explored in this paper is positioned as a way to 'explain grokking'.

Comparison with other complexity measures. The empirical results in the paper are nice. But it would be good to have a fair comparison of how other complexity measures look when measure in the same scenarios. It's unfair to say that this new complexity measure "explains" grokking without uncovering a scenario where this complexity measure is able to capture this behavior where others are not. Otherwise, it's unclear if this is just a correlation relationship with the perceived behavior of 'grokking'.

Lacking discussion of the cost for computing this complexity measure. If I understand correctly, the proposed complexity measure involves a Bayesian optimization procedure for finding the optimal compression parameters, which could be computationally expensive. It would be nice to address or (ideally) investigating how difficult this measure is. This would  enhance the practicality of the approach.

From what I understand, this complexity measure is somewhat dependent on the hyperparameters, in particular the per-layer truncation threshold $\kappa(\tau)$. It would be nice ot have a detailed analysis even experimentally of the sensitivity to this threshold.

This paper has some very nice ideas and is worth exploring but it would be good to have a section on Limitations of their approach with an honest assessment in terms of other complexity measures and the degree to which the results are not just correlational with this 'grokking' behavior.

The paper is carefully written and has a nice re-cap of the relevant ideas from information theory and compression in ML. However, the main message of the paper was at times hard to find. For example, what is the exact definition of this new complexity? I understand it relies on coarse-graning of the network and compression using bzip2 and I think the size of the compressed network is the proxy for the complexity. Is that the definition? This paper would benefit from more clear exposition in this respect.

**Questions:**

- What is the exact definition of the novel complexity measure introduced in this paper? And for which models is this measure well-dfined. The related conversation about compression and motivation from information theory and Kolmogorov complexity is very nice but it's unclear to me exactly how this measure is defined. Is this the content of Algorithm 2? Does the output of Algorithm 2 define the complexity measure?
 - in line 400, can you clarify which subset of grokking experiments you used. And why you used this subset.
 - in line 358 you state "..we show that regularizing the spectral entropy leads to grokking.." Is this an overstatement? How exactly is grokking defined quantitatively?
 - In Figure 3, you compare your regularization technique with weight decay. What is the dependence of the proposed spectral entropy regularization on the regularization weight? What behavior do you notice as you apply more or less spectral regularization? It would be nice to see the effect as the regularizaiton of the spectral entropy gradually increases.
 - Does Figure 4 include multiple seeds? Why are error bars not visible in this plot?

typos/nits
 - in Figure 2. Why include the "ours" distinction when all plots are "ours".
 - line 372, "ideas" to "ideal"

---

> ### Author Response · Authors · 2024-11-15
> **Response 1.1**
>
> Thanks for your review.
>
> **Comparison with other complexity measures**: The most commonly used proxy of complexity in the machine learning community is the L2 norm (followed by parameter count) [1,2]. As we discuss in the paper in the Related Work and elsewhere, the L2 norm is not a proper complexity metric, and its use as a complexity measure has led to a substantial amount of confusion. Indeed, one cannot construct a generalization bound from the L2 norm alone. To see this, note that a network can be re-scaled arbitrarily: We can multiply all our weights by an arbitrarily large or small constant. If a network’s weight are all 10^5 or all 10^-5, the L2 norm reports vastly different “complexities”, whereas both are in fact simple. A true measure of complexity must have units of information.
>
> On the other hand, Kolmogorov complexity is universal, and can be explicitly connected with generalization, as we show in Equation 4.
>
> In a few very specific cases of particular statistical hypothesis classes, one can construct alternative complexity measures. However, these measures simply do not apply to generic neural networks, and so have no relevance in this setting. In addition to demonstrating the complexity dynamics that occur during grokking, we are trying to clarify the situation regarding model complexity by giving an explicit upper bound on a universal complexity measure via compression.
>
> **Correlation vs Causation**: You point out that it is unclear whether our measure only “correlates” with generalization, and that seeing it compared to other measures would help. In fact, this is the point of Equation 4: it guarantees a bounding relationship between the Kolmogorov complexity and the generalization performance. In related works like [1], they cannot guarantee any generalization performance–this is because their measures are not true complexity measures.
>
> **Cost of Computing Complexity**: We use a Bayesian optimizer to search for coarse-graining settings (ways of reducing the information content of the network) which achieve the same performance as the original network. Algorithm 1 shows this procedure. The number of Bayesian optimization steps is a hyperparameter. In our experiments, we set it to 50, which is relatively modest. This parameter could be changed to support various compute budgets.
>
> In this work we are not concerned with the computational cost of the complexity estimate: the networks and datasets are not very large. The central goal is to understand the complexity dynamics in depth. The complexity estimation budget will vary depending on the goal of the work. Here we want to get a good estimate of the complexity at every step. In many cases, practitioners will only want to know the complexity of the final model, and can perform this step only one time, after training.
>
> __
>
> > From what I understand, this complexity measure is somewhat dependent on the hyperparameters, in particular the per-layer truncation threshold κ(τ)
>
> This is not correct: k(\tau) is not a hyperparameter. The Bayesian optimization procedure searches for values of k(\tau) at each step, to produce the tightest complexity estimate within its compute budget, as mentioned above. k(\tau) is merely a way to allow for different degrees of rank decomposition per layer.
>
> > What is the exact definition of the novel complexity measure introduced in this paper?
>
> The measure of complexity we use is given in Equation 5, the algorithmic rate–distortion function. It returns the Kolmogorov complexity of the coarse model which satisfies the distortion bound.
>
> > For which models is this complexity measure well defined?
>
> This complexity measure is well-defined for all possible models. Kolmogorov complexity is universal.
>
> > in line 400, can you clarify which subset of grokking experiments you used. And why you used this subset.
>
> We have expanded the discussion in the experiments section. For simplicity, we chose the first 4 grokking experiments reported in the original grokking paper [3], and which are also studied in [1]. These tasks are the best studied, but we had no other particular reason for choosing them.
>
> We view this simplicity as a virtue: complexity is a difficult and subtle concept, as this discussion shows. We choose to study complexity in the simplest possible setting to clarify its nature. The original grokking paper [3] studies 12 different simple algorithmic tasks made of binary operations. [1] reduce this to a subset of 9 of the original tasks, although they increase the size of the prime field from 97 to 113. We have expanded our discussion of these settings and added a hyperparameter table to the appendix to more clearly explain our experimental setup.

---

> ### Author Response · Authors · 2024-11-15
> **Response 1.2**
>
> > in line 358 you state "..we show that regularizing the spectral entropy leads to grokking.." Is this an overstatement? How exactly is grokking defined quantitatively?
>
> No, this is not an overstatement. Regularizing the spectral entropy alone does cause grokking, which is defined as perfect generalization after overfitting. It is not difficult to cause grokking with different regularizers, however, this is not a central claim of our work, so we have changed this line to better make our point: we now merely state that our regularization method also causes grokking. The final plot in the appendix shows grokking induced by our regularization method. Grokking is defined by a distinct memorization phase where train accuracy is 100%, and test accuracy is low (<30%, often 0%), followed by a generalization phase where test accuracy goes to 100%. We plotted these accuracy curves in the final figure in the appendix to demonstrate that all regularized networks grok, and unregularized networks do not grok.
>
>
> > In Figure 3, you compare your regularization technique with weight decay. What is the dependence of the proposed spectral entropy regularization on the regularization weight? What behavior do you notice as you apply more or less spectral regularization? It would be nice to see the effect as the regularizaiton of the spectral entropy gradually increases.
>
> Like any regularization method, one ideally wants to find a good hyperparameter for each new model class, loss function, and optimizer. There is no universally correct regularization weight, generally. We apologize that a hyperparameter table was missing from the original submission. We have included a hyperparameter table in the updated draft.
>
> As we show in the updated draft, Fig 8 in the appendix, the effect of the spectral entropy regularization can be seen clearly as a decrease in the effective rank of the matrix.
>
> >Does Figure 4 include multiple seeds? Why are error bars not visible in this plot?
>
> Yes, the total description length plots are produced with multiple seeds. The lack of error bars was an oversight on our part, which we have remedied in the updated draft.
>
> > in Figure 2. Why include the "ours" distinction when all plots are "ours".
>
> We have updated this figure to remove “ours”, and only mention it in the caption. We have also added the complexity and accuracy plots for unregularized networks, so that you can see the relationship of the complexity dynamics with train and test accuracy in the case where generalization does not occur.
>
> **Conclusion**: Overall, we wish to emphasize that prior works which study complexity in grokking do not use true complexity measures, only proxies of complexity. Our complexity metric is based on the Kolmogorov complexity, which is universal and so can be applied to any model class. Our use of Kolmogorov complexity results in explicit generalization bounds.
>
> We agree with your point that one wants to know whether the complexity measure being used guarantees generalization performance, vs merely correlates with generalization. This is *why* one wants an explicit generalization bound, and is a particular strength of our work vs previous works. Because they only study proxies of complexity, they cannot, in general, produce such bounds, whereas we can since we use a universal measure of complexity.
>
> We are excited to share this development in the theory of complexity and generalization with the community, and think that many people will be interested.
>
> [1]: Varma, Vikrant, Rohin Shah, Zachary Kenton, J'anos Kram'ar and Ramana Kumar. “Explaining grokking through circuit efficiency.” ArXiv abs/2309.02390 (2023)
>
> [2]: Nakkiran, Preetum, Gal Kaplun, Yamini Bansal, Tristan Yang, Boaz Barak and Ilya Sutskever. “Deep double descent: where bigger models and more data hurt.” Journal of Statistical Mechanics: Theory and Experiment 2021 (2019)
>
> [3]: Power, Alethea, Yuri Burda, Harrison Edwards, Igor Babuschkin and Vedant Misra. “Grokking: Generalization Beyond Overfitting on Small Algorithmic Datasets.” ArXiv abs/2201.02177 (2022)

---

### Meta-Review · Area_Chair_XkvX · 2024-12-19

**Metareview:**

**Summary of Discussion:**
The paper introduces a novel complexity measure to study grokking dynamics and explores its implications for regularization via spectral entropy. The reviewers were divided, with some appreciating the theoretical angle while others highlighted critical gaps.

**Key Concerns:**

1. **Experimental Scope and Generality:**
   - The experiments were confined to modulo arithmetic, limiting the generality of findings.
   - The lack of validation on other tasks where grokking has been observed undermines the universality of claims.

2. **Practical Impact of Regularization:**
   - While the proposed regularization aligns with the theoretical framework, it did not demonstrate notable practical benefits over weight decay.
   - The authors’ acknowledgment that performance gains were not a focus raised questions about the practical utility of the measure.

3. **Clarity and Theoretical Justification:**
   - Some core notions, such as capacity and distortion, lacked formal definitions, leaving room for interpretation and misunderstanding.
   - The theoretical connection to Kolmogorov complexity, while promising, would benefit from greater formal rigor and explanation.

**Conclusion:**
The work provides an intriguing perspective on grokking but falls short on broader applicability and clarity. With expanded experiments across diverse tasks, more explicit practical implications, and clearer theoretical exposition, this paper could better meet the community's expectations.

**Additional Comments On Reviewer Discussion:**

See above

---

### Decision · Program_Chairs · 2025-01-22

Reject